## RESEARCH ARTICLE

# A novel 14-3-3θ phosphomimetic mouse model demonstrates social dominance defects

Mary A. Gannon, Thanushri Srikantha, Rudradip Pattanayak, Navya Kapa, Aneesh Pathak, A. Claire Roberts, William J. Stone, Kasandra Scholz, Roschongporn Ekkatine and Talene A. Yacoubian*

## ABSTRACT

14-3-3 proteins, particularly the 14-3-3θ isoform, are neuroprotective in several models of Parkinson's disease (PD). Evidence for increased 14-3-3θ phosphorylation observed in PD and other neurodegenerative diseases points to a possible pathogenic role for 14-3-3θ phosphorylation in neurodegenerative disease. We recently created a novel conditional knock-in mouse to express the 14-3-3θ S232D phosphomimetic mutation. After crossing this conditional knock-in mouse with the Emx1-Cre mouse in order to induce expression of the S232D mutation in the cortex and hippocampus, we evaluated the effect of 14-3-3θ phosphorylation on behavior and pathology. These mice demonstrated mild motor deficits and reduced social dominance behavior but showed normal cognition and anxiety levels compared to Cre control mice. S232D mice did not show any α-synuclein or phospho-tau pathology at baseline, and dendritic arborization was normal in primary hippocampal cultures from S232D mice. Overall, this mouse model is a novel tool that can be used to look at the effect of 14-3-3θ phosphorylation at S232 in the context of neurodegenerative disease models.

KEY WORDS: 14-3-3, Alpha-synuclein, Tau, Parkinson's disease, Alzheimer's disease, Phosphorylation, Mouse models, NMDA receptors

## INTRODUCTION

The highly conserved family of 14-3-3 proteins are implicated in many cellular roles, including protein folding, protein trafficking, and inhibition of apoptosis (Giusto et al., 2021; Mrowiec and Schwappach, 2006; Pair and Yacoubian, 2021; Sluchanko and Gusev, 2017; Vincenz and Dixit, 1996; Yano et al., 2006). These proteins mediate their biological functions via protein-protein interactions (PPIs) (Dougherty and Morrison, 2004; MacKintosh, 2004; Pair and Yacoubian, 2021). Through PPIs, 14-3-3s can alter a binding partner's conformation, enzymatic activity, and/or subcellular localization (Dougherty and Morrison, 2004; MacKintosh, 2004; Pair and Yacoubian, 2021). 14-3-3s are expressed at high levels in the brain and regulate neurite growth, synaptogenesis, and axonal trafficking (Chapman et al., 2019; Cornell and Toyo-Oka, 2017; Giusto et al., 2021; Lavalley et al., 2016; Pandey and Smith, 2011; Taya et al., 2007). Alterations in 14-3-3s likely play a role in the pathological processes in neurological disease (Giusto et al., 2021; Pair and Yacoubian, 2021). In particular, 14-3-3s interact with a number of proteins implicated in neurodegenerative diseases, including α-synuclein (αsyn), parkin, LRRK2, beta-amyloid (Aβ), and tau, among others (Dzamko et al., 2010; Hashiguchi et al., 2000; Li et al., 2011; Liao et al., 2004; Nichols et al., 2010; Ostrerova et al., 1999; Sato et al., 2006; Sumioka et al., 2005; Umahara et al., 2004; Xu et al., 2002). Interestingly, 14-3-3 binding partners tend to be disordered, with a propensity towards aggregation, pointing to a chaperone-like function for 14-3-3s (Segal et al., 2023).

Our laboratory has shown that the 14-3-3θ isoform, in particular, is protective in several Parkinson's disease (PD) models. Overexpression of 14-3-3θ reduces toxicity in rotenone, N-methyl-4-phenyl-1,2,3,6-tetrahydropyridine (MPTP), αsyn, and LRRK2 models, while inhibition with the competitive pan inhibitor difopein exacerbates toxicity in these models (Ding et al., 2015; Lavalley et al., 2016; Slone et al., 2011; Underwood et al., 2021; Wang et al., 2018; Yacoubian et al., 2010). We have shown that 14-3-3θ has chaperone-like properties against αsyn. In both the paracrine αsyn model and the *in vitro* αsyn preformed fibril (PFF) model, we found that overexpression of 14-3-3θ reduced αsyn oligomerization, seeding, and internalization (Wang et al., 2018). In contrast, 14-3-3 inhibition with difopein potentiated αsyn seeding and cell-to-cell spreading (Wang et al., 2018). In the *in vivo* PFF model, mice overexpressing 14-3-3θ showed a delay in αsyn inclusion formation and reduced dopaminergic cell death in the substantia nigra (Underwood et al., 2021). Conversely, mice expressing difopein showed acceleration of αsyn inclusion formation and subsequent increase in cell death (Underwood et al., 2021). Taken together, these data demonstrate that 14-3-3θ is protective against PD-related pathology and subsequent cell death.

What prevents 14-3-3θ's protective effect in disease is not known. We observed an increase in 14-3-3θ phosphorylation at serine 232 (S232) in the Triton X-100 insoluble fractions from human PD, dementia with Lewy bodies (DLB), and Alzheimer's disease (AD) cortical brain lysates (McFerrin et al., 2017). Changes at other 14-3-3 phosphorylation sites were not observed in most of these disorders (McFerrin et al., 2017). Increased S232 phosphorylation inversely correlated with cognitive performance (McFerrin et al., 2017). We have also observed that S232 phosphorylation is increased in rotenone and αsyn overexpression models (Slone et al., 2015). A phosphomimetic S232D mutant was unable to protect against neurotoxins, mutant LRRK2, or αsyn, whereas a non-phosphorylatable S232A mutant was protective in these models (Pattanayak et al., 2024; Slone et al., 2015; Wang et al., 2025).

To test the impact of 14-3-3θ phosphorylation at S232 *in vivo*, we developed a novel conditional knock-in (KI) S232 14-3-3θ

University of Alabama at Birmingham, Center for Neurodegeneration and Experimental Therapeutics, Department of Neurology, Birmingham, AL 35294, USA.

*Author for correspondence (tyacoubian@uabmc.edu)

M.A.G., 0000-0001-9591-9711; T.A.Y., 0000-0003-2227-7310

Biology Open

phosphomimetic mouse that expresses the mutant S232D upon Cre recombination (Wang et al., 2025). We crossed this mouse with an Emx1-Cre mouse (Gorski et al., 2002) in order to express the phosphomimetic mutation in the cortex and hippocampus, brain areas affected in DLB, AD, and PD. Here, we describe the behavioral and pathologic analyses of these mice at several time points.

## RESULTS

We created a conditional KI mouse targeting the 14-3-3θ gene (*YWHAQ*), such that serine 232 (S232) was changed to an aspartic acid (S232D) upon Cre recombination, as previously described (Wang et al., 2025). Mice heterozygous for the mutant allele were bred to produce mice that were homozygous for S232D. These mice were then crossed with heterozygous Emx1-Cre mice, leading to 50% of the resultant offspring that were $Emx^{Cre/+}$ $S232^{WT/D}$. In order to generate experimental and control littermates, $Emx^{Cre/+}$ $S232^{WT/D}$ mice were then crossed with $Emx^{+/+}$ $S232^{WT/D}$ mice. From this cross, we generated $Emx^{+/+}$ $S232^{WT/WT}$ (WT), $Emx^{Cre/+}$ $S232^{WT/WT}$ (Cre control), and $Emx^{Cre/+}$ $S232^{D/D}$ (homozygous S232D) mice that were used for behavioral studies. We confirmed RNA expression for the S232D mutant in the cortex of the homozygous S232D mice by sequencing (Fig. S1A,B). Qualitatively, we also looked at 14-3-3θ expression by immunohistochemistry in the cortex and hippocampus of Cre control ($Emx^{Cre/+}$ $S232^{WT/WT}$) and homozygous S232D ($Emx^{Cre/+}$ $S232^{D/D}$) mice. Immunostaining for 14-3-3θ revealed no obvious differences in expression in either region between Cre control ($Emx^{Cre/+}$ $S232^{WT/WT}$) and homozygous S232D ($Emx^{Cre/+}$ $S232^{D/D}$) mice (Fig. S1C).

We performed motor and cognitive behavioral tests in control and $Emx^{Cre/+}$ $S232^{D/D}$ mice (referred to as S232D mice from hereon) across two separate cohorts at multiple time points. In the first cohort, we tested two types of control mice, the $Emx^{+/+}$ $S232^{WT/WT}$ mice (referred to as WT mice from hereon) and $Emx^{Cre/+}$ $S232^{WT/WT}$ mice (referred to as Cre control mice from hereon). In the second cohort, we used the Cre control mice only. The S232D and Cre control mice from both cohorts were pooled in the behavioral tests described below.

### S232D mice exhibit mild motor impairment

Motor deficits have been described in several PD mouse models. We first compared the S232D and Cre control mice in a small battery of motor tests at three different time points: 3 months, 9 months, and 12 months of age. In the open field test, the mice displayed no difference in distance traveled at any of the time points [Fig. 1A, 3 month: unpaired, two-tailed *t*-test: t (37)=0.2847, *P*=0.7774; 9 month: unpaired, two-tailed *t*-test: t (37)=0.1533, *P*=0.8790; 12 month: unpaired, two-tailed *t*-test: t (34)=1.297, *P*=0.2035]. Similarly, the S232D and Cre control mice from cohort 2 showed no significant difference in the grip strength test [Fig. 1B, 3 month: unpaired, two-tailed *t*-test: t (19)=0.09110, *P*=0.9284; 9 month: unpaired, two-tailed *t*-test: t (19)=0.8946, *P*=0.3822; 12 month: unpaired, two-tailed *t*-test: t (19)=1.220, *P*=0.2374]. Since different force grip instruments were used for mice in cohort 1 vs cohort 2, we were unable to pool data from both cohorts together. However, cohort 1 similarly did not show any significant difference in grip strength between Cre control and S232D mice [Fig. S2, 3 month: unpaired, two-tailed *t*-test: t (17)=1.664, *P*=0.1143; 9 month: unpaired, two-tailed *t*-test: t (17)=0.4698, *P*=0.6444; 12 month: unpaired, two-tailed *t*-test: t (14)=0.3825, *P*=0.7079].

We next ran the mice on the rotor rod and saw minimal differences in the learning curve across the five training and testing days. While both the Cre control and S232D mice improved in their ability to stay on the rotating rod over time, there was a significant effect of the interaction between the genotype and test day at the 9 month time point only, indicating a change in the rate of improvement between the groups, with the Cre control mice showing more improvement than the S232D mice [Fig. 1C, 3 month: two-way repeated ANOVA: genotype *F* (1, 37)=1.738, *P*=0.1955, testing day *F* (2.015, 74.57)=24.96, *P*<0.0001, interaction *F* (4, 148)=0.9262, *P*=0.4505; 9 month: two-way repeated ANOVA: genotype *F* (1, 37)=3.451, *P*=0.0712, testing day *F* (2.586, 95.69)=15.18, *P*<0.0001, interaction *F* (4, 148)=2.845, *P*=0.0262; 12 month: two-way repeated ANOVA: genotype *F* (1, 34)=0.7792, *P*=0.3836, testing day *F* (2.169, 73.75)=6.426, *P*=0.0021, interaction *F* (4, 136)=0.4311, *P*=0.7860]. Additionally, there was no difference between groups on the assessment days (average of two testing days, three trials per day averaged) at any of the time points [Fig. 1D, 3 month: unpaired, two-tailed *t*-test: t (37)=1.270, *P*=0.2121; 9 month: unpaired, two-tailed *t*-test: t (37)=1.910, *P*=0.0639; 12 month: unpaired, two-tailed *t*-test: t (34)=0.2609, *P*=0.7958].

We did, however, find that the S232D mice did significantly worse on the four-paw wire hang task. While we did not see a difference at the early 3 month time point [Fig. 2A, unpaired, two-tailed *t*-test: t (37)=1.738, *P*=0.0906], at both 9 and 12 months the S232D mice held on to the apparatus for significantly less time than their Cre control counterparts [Fig. 2A, 9 month: unpaired, two-tailed *t*-test: t (37)=2.334, *P*=0.0251; 12 month: unpaired, two-tailed *t*-test: t (34)=2.183, *P*=0.0360]. Since wire hang can be affected by weight, we corrected the wire hang testing for potential weight differences between the Cre control and S232D mice. The S232D mice were slightly heavier than the Cre controls at both 3 and 9 months but not at 12 months [Fig. 2B, 3 month: unpaired, two-tailed *t*-test: t (37)=2.033, *P*=0.0492; 9 month: unpaired, two-tailed *t*-test: t (37)=2.443, *P*=0.0194; 12 month: unpaired, two-tailed *t*-test: t (34)=2.021, *P*=0.0512]. In order to take the weight of the mice into account with the wire hang data, we calculated the holding impulse (HI) by taking the weight of the mice in grams and multiplying that by the hang time in seconds (Hoffman and Winder, 2016). The HI for the S232D mice was still significantly reduced at both 9 and 12 months compared to Cre control mice [Fig. 2C, 3 month: unpaired, two-tailed *t*-test: t (37)=1.332, *P*=0.1911; 9 month: unpaired, two-tailed *t*-test: t (37)=2.437, *P*=0.0198; 12 month: unpaired, two-tailed *t*-test: t (34)=2.057, *P*=0.0474]. Of note, at the 3 month time point, the first cohort of mice that were run on the wire hang had a maximum time of 3 min before the test was ended. A true difference may have been masked by capping the maximum time, as many of the mice in both groups reached it. Further, the HI is only accurate with unlimited hang time. However, the results were similar when only cohort 2, where the mice were given unlimited hang time, was considered.

### S232D mice show impairments in social dominance

Impairment in the social dominance test has been observed in certain neurodegenerative mouse models, including the αsyn preformed fibril (PFF) model (Stoyka et al., 2020; Underwood et al., 2021). This test examines the integrity of the prefrontal-amygdala circuitry that is affected in DLB and other neurodegenerative disorders (Arrant et al., 2016; Miczek et al., 1974; Soumiya et al., 2016; Wallén-MacKenzie et al., 2009; Zhou et al., 2018). Notably, we found a deficit in social dominance behavior in S232D mice at all three time points tested. Social dominance was run only on the first cohort of mice. At the initial 3 month time point, we tested three different sets of pairings: Cre control vs WT, S232D vs WT, and S232D vs Cre control. While

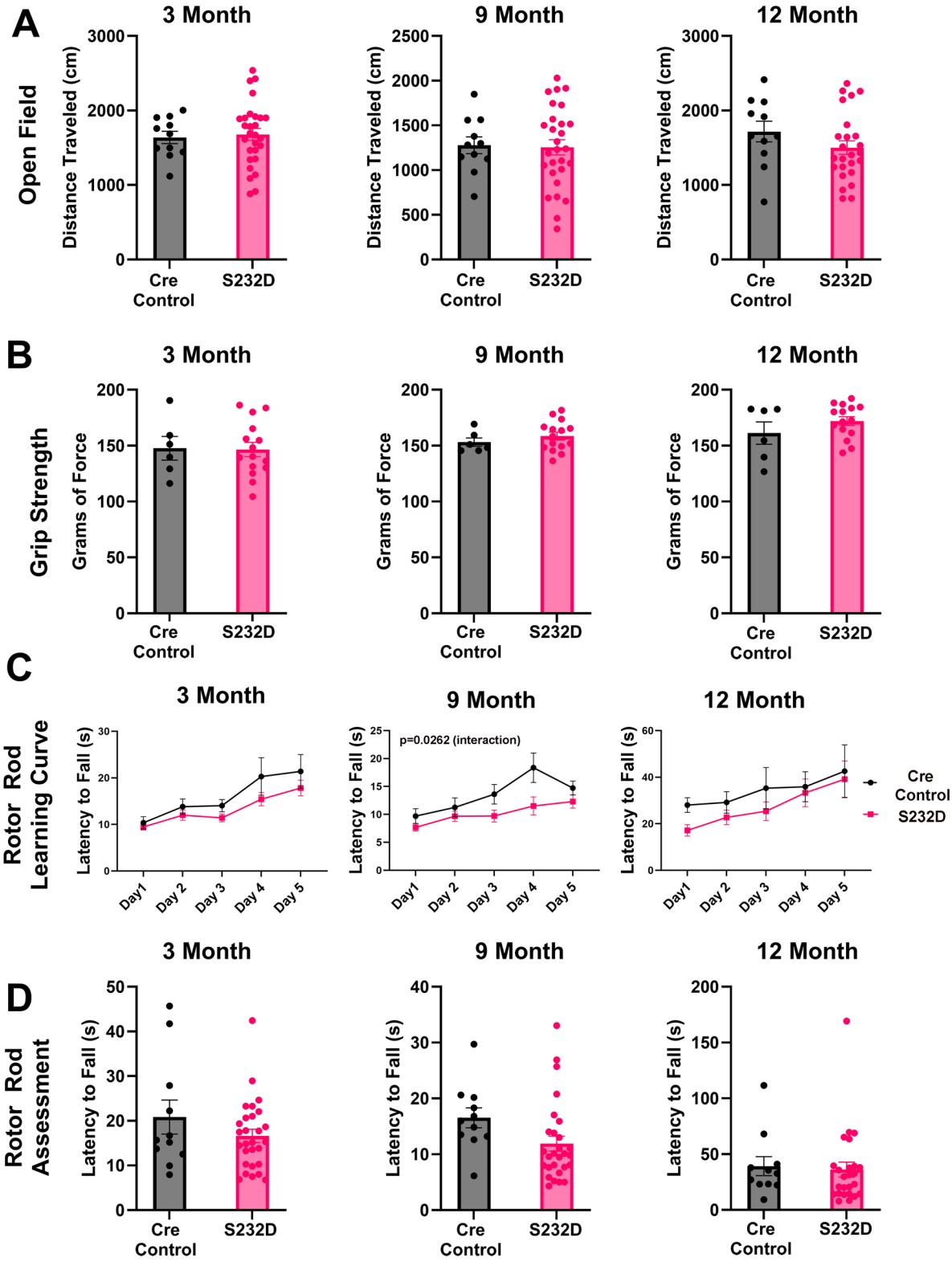

**Fig. 1. S232D mice do not demonstrate deficits in open field, grip strength, or rotor rod tests.** (A) Quantification of distance traveled in the open field task at 3, 9, and 12 months. $n$=11-28 mice per group. Error bars represent s.e.m. (B) Quantification of grip strength at 3, 9, and 12 months. $n$=6-15 mice per group. Error bars represent s.e.m. (C) Quantification of latency to fall in rotor rod at 3, 9, and 12 months. $n$=11-28 mice per group. Error bars represent s.e.m. (D) Quantification of average latency to fall on the two test days at 3, 9, and 12 months. $n$=11-28 mice per group. Error bars represent s.e.m.

we saw no difference in win percentage between WT and Cre control groups [Fig. 3B, two-tailed $t$-test: t (15)=0.4403, $P$=0.6660], the S232D mice lost significantly more to the Cre control mice [Fig. 3A, two-tailed $t$-test: t (16)=4.022, $P$=0.0010] and also neared significance in losing more to the WT mice [Fig. 3C, two-tailed $t$-test: t (21)=2.036, $P$=0.0545]. Since the WT and Cre control mice

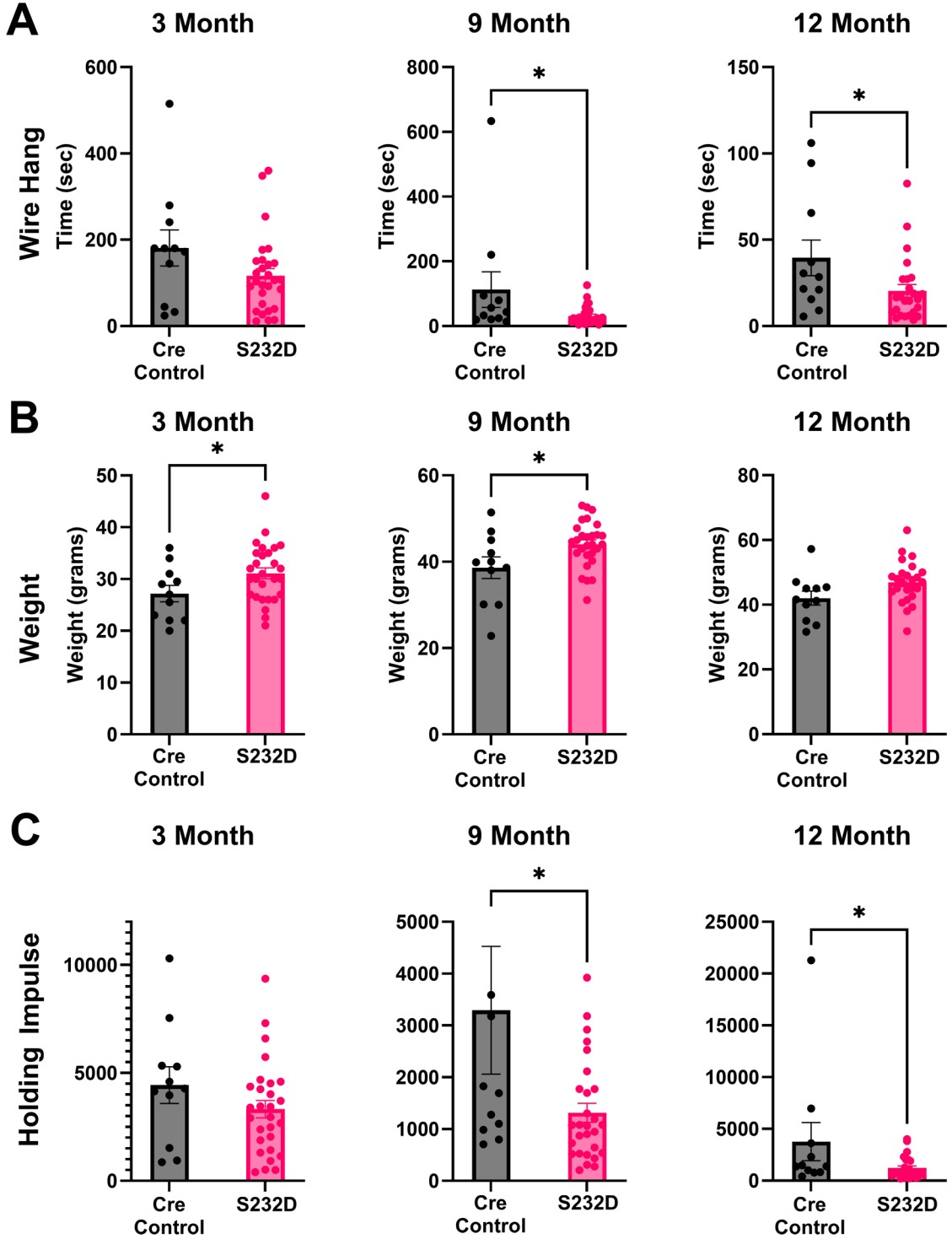

**Fig. 2. S232D mice are impaired in four-paw wire hang.** (A) Quantification of latency to fall from wire hang apparatus at 3, 9, and 12 months. *n*=11-28 mice per group. *$P \leq 0.05$ (unpaired, two-tailed *t*-test). Error bars represent s.e.m. (B) Mouse weights at 3, 9, and 12 months. *n*=11-28 mice per group. *$P \leq 0.05$ (unpaired, two-tailed *t*-test). Error bars represent s.e.m. (C) Holding impulse, calculated as weight in grams multiplied by the hang time in seconds, at 3, 9, and 12 months. *n*=11-28 mice per group. *$P \leq 0.05$ (unpaired, two-tailed *t*-test). Error bars represent s.e.m.

performed similarly at 3 months, and due to a limited number of Cre control mice, we used the WT mice to compare to the S232D mice at the remaining two time points. S232D mice won significantly less than WT mice at 9 months [Fig. 3D, two-tailed *t*-test: t (21)=4.518, *P*=0.0002] and also at 12 months [Fig. 3E, two-tailed *t*-test: t (20)=4.871, *P*<0.0001].

**S232D mice show no abnormalities in anxiety or cognitive testing**

Anxiety and cognitive impairment are common deficits observed in DLB and AD mouse models. We used the elevated plus/zero maze to test anxiety levels in the S232D mice. The S232D mice spent a similar amount of time in the open arm of the elevated plus maze

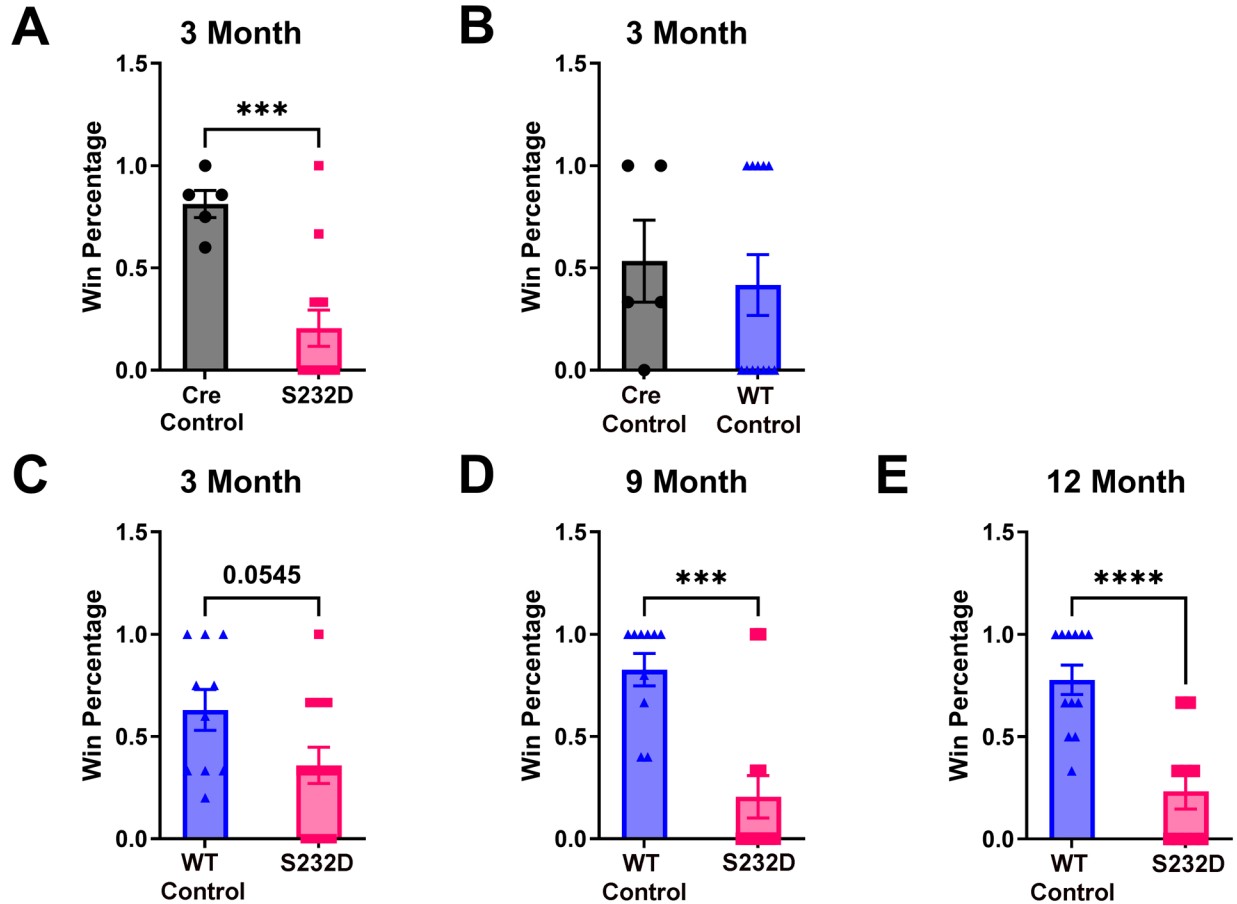

**Fig. 3. S232D mice show impairments in social dominance**. (A) Quantification of tube test win rate of S232D mice matched against Cre control mice at 3 months of age. *n*=5-14 mice per group. ***P≤0.001 (unpaired, two-tailed *t*-test). Error bars represent s.e.m. (B) Quantification of tube test win rate of WT control mice matched against Cre control mice at 3 months of age. *n*=5-12 mice per group. Error bars represent s.e.m. (C) Quantification of tube test win rate of S232D mice matched against WT control mice at 3 months of age. *n*=10-13 mice per group. *P*=0.054 (unpaired, two-tailed *t*-test). Error bars represent s.e.m. (D) Quantification of tube test win rate of S232D mice matched against WT control mice at 9 months of age. *n*=10-13 mice per group. ***P≤0.001 (unpaired, two-tailed *t*-test). Error bars represent s.e.m. (E) Quantification of tube test win rate of S232D mice matched against WT control mice at 12 months of age. *n*=10-12 mice per group. ****P≤0.0001 (unpaired, two-tailed *t*-test). Error bars represent s.e.m.

(EPM; 3 months) or elevated zero maze (EZM; 9 and 12 months) as the Cre control mice, indicating no change in levels of anxiety for the S232D mice [Fig. 4A, 3 month: unpaired, two-tailed *t*-test: t (16)=0.6828, *P*=0.5045; 9 month: unpaired, two-tailed *t*-test: t (16)=0.4877, *P*=0.6324; 12 month: unpaired, two-tailed *t*-test: t (13)=0.5872, *P*=0.5671].

We similarly saw minimal changes in the performance of S232D mice in the Morris water maze. Across the five days of training, the S232D mice were slightly faster at finding the location of the hidden platform at 12 months, but the learning curves showed no significant differences at either 3 or 9 months of age [Fig. 4C, 3 month: two-way repeated ANOVA: genotype *F* (1, 16)=0.5092, *P*=0.4858, testing day *F* (2.743, 43.88)=20.15, *P*<0.0001, interaction *F* (4, 64)=1.126, *P*=0.3524; 9 month: two-way repeated ANOVA: genotype *F* (1, 16)=0.07886, *P*=0.7824, testing day *F* (2.841, 45.45)=3.062, *P*=0.0399, interaction *F* (4, 64)=0.4092, *P*=0.8014; 12 month: two-way repeated ANOVA: genotype *F* (1, 13)=7.489, *P*=0.0170, testing day *F* (1.734, 22.54)=10.79, *P*=0.0008, interaction *F* (4, 52)=0.6868, *P*=0.6044]. In the probe trial, the S232D mice and Cre control mice showed no significant difference in the percent time spent in the target quadrant at any of the time points [Fig. 4D, 3 month: unpaired, two-tailed *t*-test: t (16)=0.9050, *P*=0.3789; 9 month: unpaired, two-tailed *t*-test: t (16)=0.3654,

*P*=0.7196; 12 month: unpaired, two-tailed *t*-test: t (13)=0.1407, *P*=0.8902).

Finally, we looked at the passive avoidance test in the S232D and Cre control mice. This test was performed only at the final 12 month time point so that the exposure to the stress of a foot shock did impact any other behavioral tests that were performed. The S232D and Cre control mice did not display a significant difference in the latency to crossing to the dark side on day 2 of the test [Fig. 4B, unpaired, two-tailed *t*-test: t (34)=0.9976, *P*=0.3255], indicating a similar ability to remember the previous day's foot shock and suppress their innate impulse to move toward the dark chamber.

## S232D mice do not show evidence of αsyn or tau pathology

Following the completion of the 12 month behavior tests, mice were sacrificed, and brains were collected to measure αsyn and pS129 phosphorylated αsyn (psyn) levels in the cortex via immunohistochemistry. Due to a limited number of Cre control mice in the initial cohort, which we used for this experiment, we used WT mice instead of Cre control mice to compare to the S232D mice. We found no difference in the intensity of αsyn immunoreactivity in the cortex [Fig. 5A, unpaired, two-tailed *t*-test: t (20)=0.4221, *P*=0.6775]. We also did a proteinase K (PK) digestion of the tissue prior to αsyn staining to compare αsyn

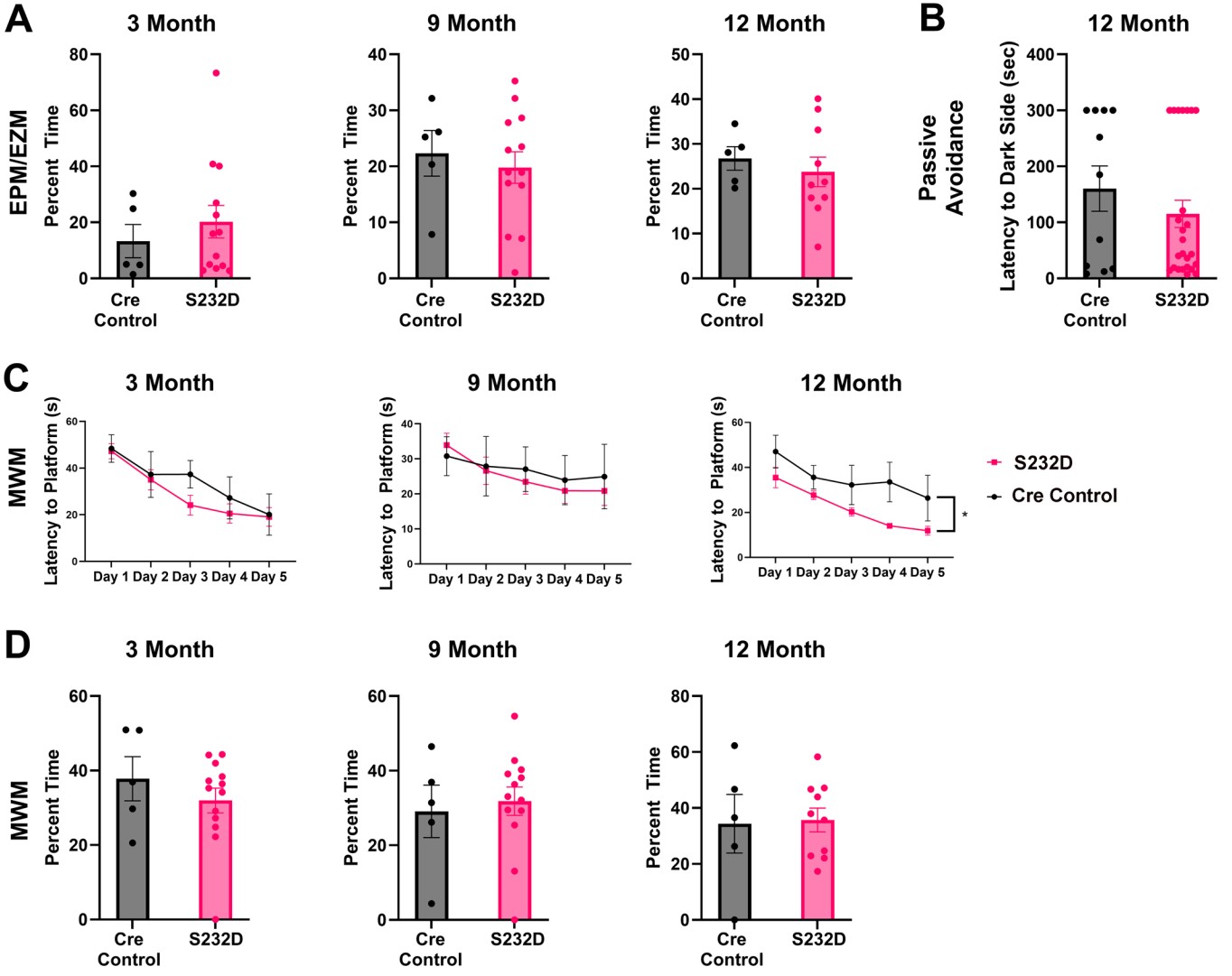

**Fig. 4. S232D mice do not display anxiety or cognitive deficits.** (A) Quantification of time spent in the open arm in the elevated plus maze (EPM) or elevated zero maze (EZM) at 3, 9, and 12 months. *n*=5-13 mice per group. Error bars represent s.e.m. (B) Quantification of latency in crossing to the dark side on day 2 of the passive avoidance task at 12 months. *n*=11-25 mice per group. Error bars represent s.e.m. (C) Quantification of latency to platform in the Morris water maze (MWM) at 3, 9, and 12 months. *n*=5-13 mice per group. **P≤0.05 (two-way repeated ANOVA). Error bars represent s.e.m. (D) Quantification of the probe trial in the MWM at 3, 9, and 12 months. *n*=5-13 mice per group. Error bars represent s.e.m.

solubility in the S232D and WT mice. We found that the αsyn intensity following PK digestion was unchanged between the two groups in cortex [Fig. 5A, unpaired, two-tailed *t*-test: t (20)=0.8092, *P*=0.4279]. Not surprisingly, with no differences between either pre or post PK digestion αsyn levels, there was similarly not a difference between the ratio of the αsyn intensity of the digested over undigested tissue [Fig. 5A, unpaired, two-tailed *t*-test: t (20)=0.4413, *P*=0.6637].

Qualitatively, we did not observe any psyn immunoreactivity in either the Cre control or S232D mice (Fig. S3; *n*=4 mice per group). To ensure that the lack of signal was not due to experimental issues, we included tissue from a mouse that had been injected with αsyn PFFs. This positive control mouse exhibited the expected psyn inclusion pattern (Fig. S3).

We also examined cortical αsyn protein levels via Western blot. Brains from 3- and 12-month-old WT (3 months only), Cre control, and S232D mice were collected and fractionated into Triton X-100 soluble and insoluble fractions. At 3 months, no differences were seen between groups in either the Triton soluble [Fig. 5C, one-way

ANOVA: *F* (2, 6)=2.278, *P*=0.1837] or insoluble [Fig. 5D, one-way ANOVA: *F* (2, 6)=1.160, *P*=0.3751] fractions. At 12 months, there was similarly no difference in αsyn levels between the Cre control and S232D in either the Triton soluble [Fig. 5E, unpaired, two-tailed *t*-test: t (6)=1.223, *P*=0.2673] or insoluble [Fig. 5F, unpaired, two-tailed *t*-test: t (6)=0.5495, *P*=0.6025] fractions.

Given that levels of 14-3-3θ phosphorylation at S232 were increased in human AD tissue as well as PD, we next examined levels of phospho-tau (ptau) at threonine 181 (ptau181) in the 12 month lysates. In both the Triton soluble and insoluble fractions, ptau181 levels were not significantly changed between the Cre control and S232D mice [Fig. 5E, Triton soluble: unpaired, two-tailed *t*-test: t (5)=1.746, *P*=0.1413; Fig. 5F, Triton insoluble: unpaired, two-tailed *t*-test: t (6)=0.2467, *P*=0.8133].

## 14-3-3θ levels are decreased in aged S232 mice
We also looked at the expression levels of 14-3-3θ in these same S232D and Cre control Triton soluble and insoluble lysates from the cortex. At 3 months, the 14-3-3θ levels were unchanged

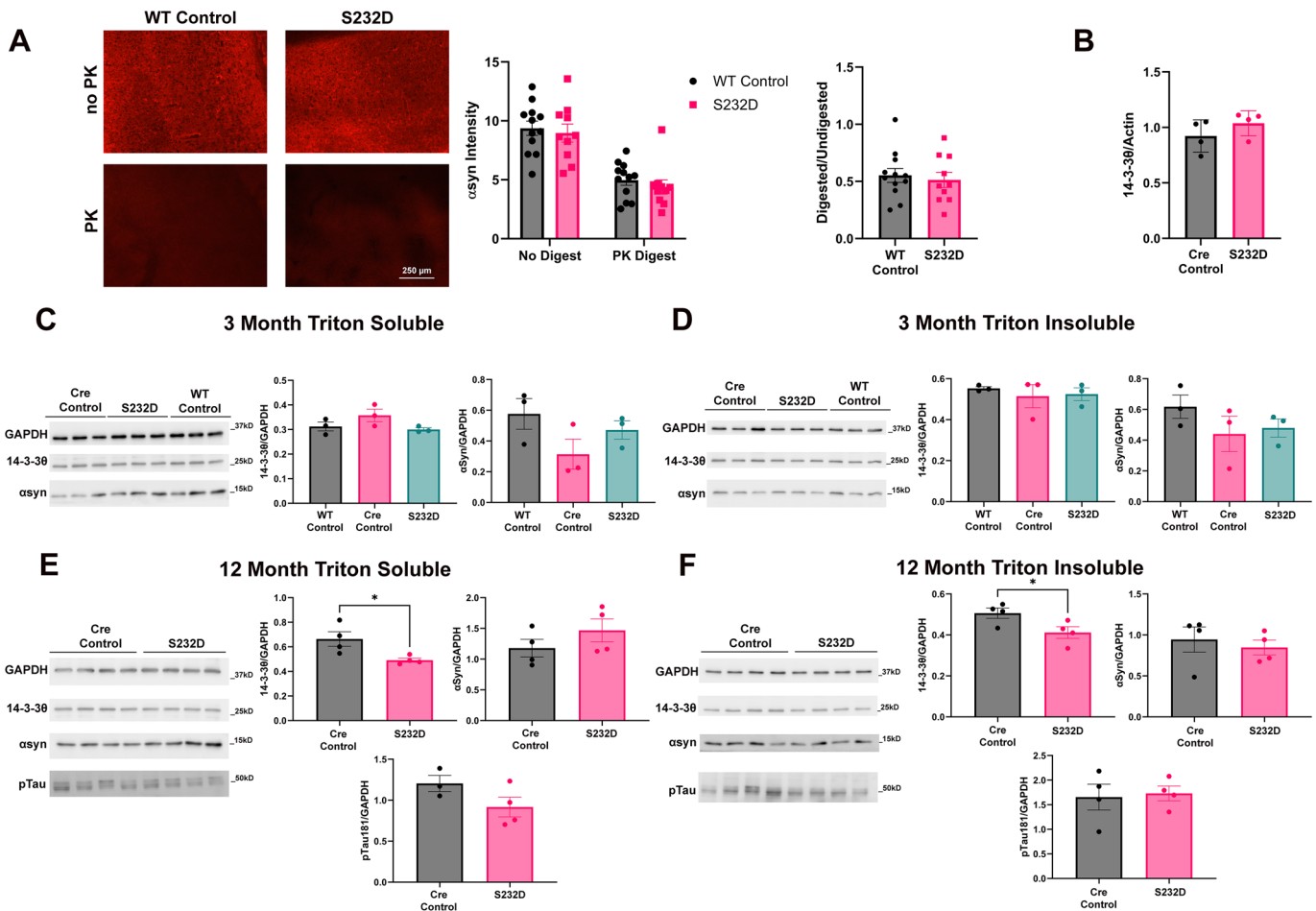

**Fig. 5. αSyn and phospho-tau levels are not altered in S232D mice.** (A) Representative images and quantification of αsyn immunostaining of S232D and WT control cortical tissue with and without digestion with proteinase K (PK). *n*=10-12 mice per group. Error bars represent s.e.m. (B) Quantification of mRNA levels of 14-3-3θ normalized to actin in the cortices of 12 month old S232D and Cre control mice. *n*=4 mice per group. Error bars represent s.e.m. (C) Representative Western blot and quantification of αsyn and 14-3-3θ levels in Triton X-100 soluble cortical fractions from 3-month-old S232D, Cre control, and WT control brains. *n*=3 mice per group. Error bars represent s.e.m. (D) Representative Western blot and quantification of αsyn and 14-3-3θ levels in Triton X-100 insoluble cortical fractions from 3-month-old S232D, Cre control, and WT control brains. *n*=3 mice per group. Error bars represent s.e.m. (E) Representative Western blot and quantification of αsyn, 14-3-3θ, and ptau levels in Triton X-100 soluble cortical fractions from 12-month-old S232D and Cre control brains. *n*=4 mice per group. *$P \leq 0.05$ (unpaired, two-tailed *t*-test). Error bars represent s.e.m. (F) Representative Western blot and quantification of αsyn, 14-3-3θ, and ptau levels in Triton X-100 insoluble cortical fractions from 12-month-old S232D and Cre control brains. *n*=4 mice per group. *$P \leq 0.05$ (unpaired, two-tailed *t*-test). Error bars represent s.e.m.

between groups in the Triton soluble [Fig. 5C, one-way ANOVA: $F_{(2, 6)}=2.683$, $P=0.1471$] and insoluble [Fig. 5D, one-way ANOVA: $F_{(2, 6)}=0.2741$, $P=0.7693$] fractions. However, at the 12 month time point, there was a small but significant decrease in 14-3-3θ levels of the S232D mice compared to the Cre control in both the Triton soluble [Fig. 5E, unpaired, two-tailed *t*-test: $t_{(6)}=2.807$, $P=0.0309$] and insoluble [Fig. 5F, unpaired, two-tailed *t*-test: $t_{(6)}=2.473$, $P=0.0482$] fractions.

Given this reduction in 14-3-3θ protein levels in the older S232D mice, we next measured the levels of 14-3-3θ RNA extracted from cortices of 12 month S232D and Cre control mouse brains using quantitative PCR (qPCR). 14-3-3θ mRNA levels were not different between S232D and Cre control cortices [Fig. 5B, unpaired, two-tailed *t*-test: $t_{(6)}=1.246$, $P=0.2591$].

## Neurons from S232D mice have normal dendritic architecture

Given the role of 14-3-3 proteins in neurite growth (Cornell and Toyo-Oka, 2017; Giusto et al., 2021), we next examined potential

structural changes to dendrites in the S232D mice compared to Cre controls. Using primary cultured hippocampal neurons (Fig. 6A), we first looked at the average neurite length and found no difference between the groups [Fig. 6B, unpaired, two-tailed *t*-test: $t_{(64)}=0.8925$, $P=0.3755$]. Similarly, there were no differences in the number of primary dendrites [Fig. 6C, unpaired, two-tailed *t*-test: $t_{(64)}=1.127$, $P=0.2640$], number of dendritic nodes [Fig. 6D, unpaired, two-tailed *t*-test: $t_{(62)}=0.04028$, $P=0.9680$], or number of dendritic terminal ends [Fig. 6E, unpaired, two-tailed *t*-test: $t_{(64)}=1.069$, $P=0.2891$]. We also performed Sholl analysis on these neurons to look at dendritic arborization (Fig. 6F) and saw no difference between the groups [Fig. 6G, unpaired, two-tailed *t*-test: $t_{(64)}=1.134$, $P=0.2612$].

## Investigation of NMDA receptor levels in the cortex of S232D mice

14-3-3 proteins regulate *N*-methyl-D-aspartate receptor (NMDAR) trafficking in neurons (Chung et al., 2015), and changes in NMDAR postsynaptic density (PSD) have been previously reported in 14-3-3

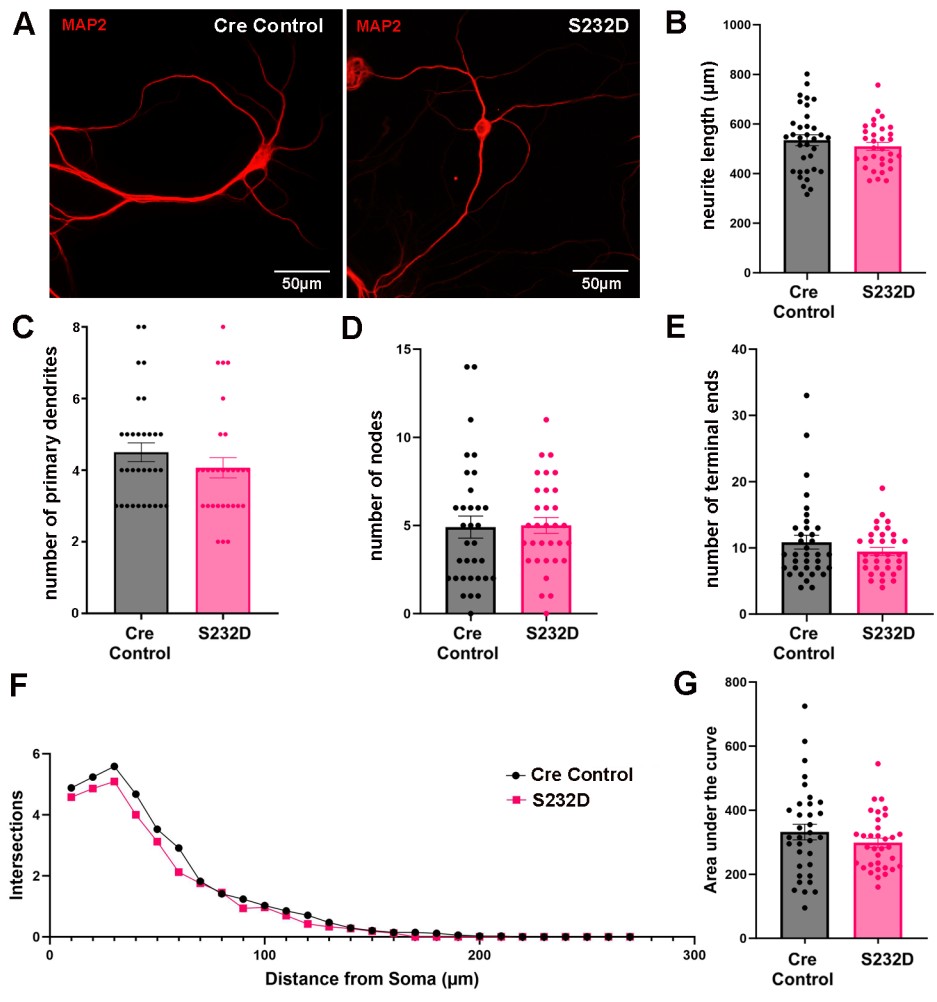

**Fig. 6. Dendritic architecture is unchanged in neurons from S232D mice.**
(A) Representative images of MAP2 immunostaining of S232D and Cre control primary neurons at DIV 8. (B) Quantification of neurite length of S232D and Cre control primary neurons at DIV 8. $n=3$ independent rounds with 10-12 neurons per group per round. Error bars represent s.e.m. (C) Quantification of number of primary dendrites of S232D and Cre control primary neurons at DIV 8. $n=3$ independent rounds with 10-12 neurons per group per round. Error bars represent s.e.m. (D) Quantification of number of nodes of S232D and Cre control primary neurons at DIV 8. $n=3$ independent rounds with 10-12 neurons per group per round. Error bars represent s.e.m. (E) Quantification of number of terminal ends of S232D and Cre control primary neurons at DIV 8. $n=3$ independent rounds with 10-12 neurons per group per round. Error bars represent s.e.m. (F) Scholl analysis of S232D and Cre control primary neurons at DIV 8. $n=3$ independent rounds with 10-12 neurons per group per round. (G) Quantification of the area under the curve in the Scholl analysis of S232D and Cre primary neurons. $n=3$ independent rounds with 10-12 neurons per group per round. Error bars represent s.e.m.

functional knockout mice (Qiao et al., 2014). To test if there were any changes in these receptors in the S232D mouse, we performed subcellular fractionation to isolate postsynaptic densities from the cortex of Cre control and S232D mice and then performed Western blot analysis of NMDAR1, 2A, and 2B receptor subunits (Fig. 7A). While we observed no changes between Cre control and S232D mice in NMDAR1 [Fig. 7B unpaired, two-tailed $t$-test: t (10)=0.7225, $P$=0.4865], we did see changes in both NMDAR2A and NMDAR2B receptor subunits when normalized to PSD95. NMDAR2A/PSD95 levels were increased in the PSD in S232D mice compared to Cre control [Fig. 7C, unpaired, two-tailed $t$-test: t (10)=5.107, $P$=0.0005]. We similarly saw an increase in NMDAR2B/PSD95 levels in S232D mice compared to Cre controls [Fig. 7D, unpaired, two-tailed $t$-test: t (10)=4.654, $P$=0.0009].

We chose to use PSD95 as the loading control in order to control for the purity of the PSD fraction, but we next tested whether there were changes in PSD95 between groups that could be affecting our results. When normalized to total protein, the PSD95 levels were reduced in the S232D mice compared to the Cre control, although not quite reaching significance [Fig. 7E, unpaired, two-tailed $t$-test: t (10)=2.079, $P$=0.0644]. Due to the differences seen in PSD95, we also normalized the NMDAR subunits to total protein. There were no significant changes between groups in NMDAR1/total protein [Fig. 7F, unpaired, two-tailed $t$-test: t (10)=1.535, $P$=0.1558], NMDAR2A/total protein [Fig. 7G, unpaired, two-tailed $t$-test: t (10)=0.6835, $P$=0.5098], or NMDAR2B/total protein [Fig. 7H, unpaired, two-tailed $t$-test: t (10)=0.4740, $P$=0.6457].

## DISCUSSION

Our previous work has shown aberrant levels of 14-3-3θ phosphorylation at S232 in several neurodegenerative diseases, including PD, DLB, and AD (McFerrin et al., 2017). Additionally, we have found that the phosphomimetic S232D 14-3-3θ mutant fails to protect against toxicity in several PD models (Pattanayak et al., 2024; Slone et al., 2015; Wang et al., 2025). Based on these findings, we developed a novel S232D mouse model in order to understand the significance of 14-3-3θ phosphorylation in the brain. Here, we describe our behavioral and pathological characterization of this S232D mouse. We found that the S232D mice demonstrate a subtle motor deficit with age in the wire hang test. Additionally, S232D mice showed deficits in social dominance. S232D mice were comparable to control mice in most motor, anxiety, and cognitive tests. These mice did not demonstrate αsyn or tau pathology at baseline, and dendritic architecture was normal in primary neuronal cultures. Based on these data, we conclude that 14-3-3θ phosphorylation induces mild behavioral deficits.

These new phosphomimetic mice will be useful in understanding the role of 14-3-3 proteins in neurodegenerative diseases. Our group has shown alterations in 14-3-3θ phosphorylation at S232 in PD, AD, and DLB (McFerrin et al., 2017), and pesticides associated with increased risk of PD can increase 14-3-3θ phosphorylation (Slone et al., 2015). This mouse will be informative for examination of downstream pathways that may be affected by S232 phosphorylation. Our laboratory has already shown that S232 phosphorylation interferes with 14-3-3θ's ability to modulate both

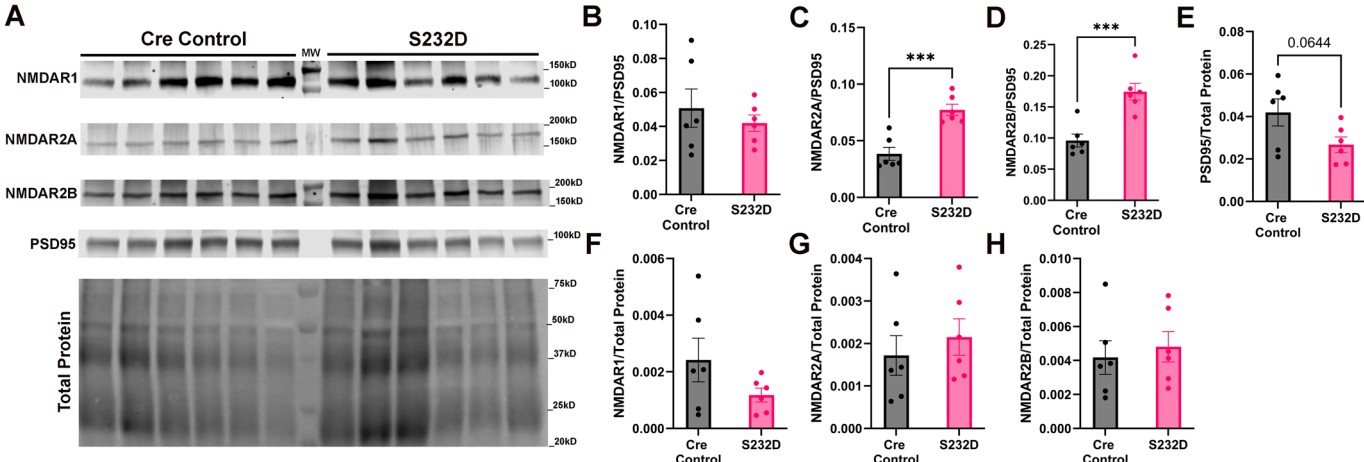

**Fig. 7. NMDA receptor levels in the cortex**. (A) Representative Western blot of NMDAR1, NMDAR2A, NMDAR2B, PSD95, and total protein in post-synaptic density (PSD) fractions from the cortices of 3-month-old S232D and Cre control brains. *n*=6 mice per group. (B) Quantification of NMDAR1 normalized to PSD95 in PSD fractions from the cortices of 3-month-old S232D and Cre control brains. *n*=6 mice per group. Error bars represent s.e.m. (C) Quantification of NMDAR2A normalized to PSD95 in PSD fractions from the cortices of 3-month-old S232D and Cre control brains. *n*=6 mice per group. ***$P{\leq}0.001$ (unpaired, two-tailed *t*-test). Error bars represent s.e.m. (D) Quantification of NMDAR2B normalized to PSD95 in PSD fractions from the cortices of 3-month-old S232D and Cre control brains. *n*=6 mice per group. ***$P{\leq}0.001$ (unpaired, two-tailed *t*-test). Error bars represent s.e.m. (E) Quantification of PSD95 normalized to total protein in PSD fractions from the cortices of 3-month-old S232D and Cre control brains. *n*=6 mice per group. Error bars represent s.e.m. (F) Quantification of NMDAR1 normalized to total protein in PSD fractions from the cortices of 3-month-old S232D and Cre control brains. *n*=6 mice per group. Error bars represent s.e.m. (G) Quantification of NMDAR2A normalized to total protein in PSD fractions from the cortices of 3-month-old S232D and Cre control brains. *n*=6 mice per group. Error bars represent s.e.m. (H) Quantification of NMDAR2B normalized to total protein in PSD fractions from the cortices of 3-month-old S232D and Cre control brains. *n*=6 mice per group. Error bars represent s.e.m.

the propagation and toxicity of αsyn (Wang et al., 2025) as well as LRRK2 kinase activity (Pattanayak et al., 2024). Additionally, molecular modeling has shown that S232 phosphorylation induces structural changes in the canonical binding pocket of 14-3-3θ (Pattanayak et al., 2024), thereby having a potential impact on many other 14-3-3θ binding partners. Current work in our laboratory is examining the global impact of this phosphorylation change on binding interactions using unbiased affinity-purified mass spectrometry.

While 14-3-3θ phosphorylation changes have been observed in PD, DLB, and AD brains (McFerrin et al., 2017), we did not observe any clear αsyn or tau pathology in the S232D mouse brains. αSyn levels were unchanged in Triton soluble and insoluble fractions, and no differences in αsyn sensitivity to proteinase K digestion were observed in S232D mice. Similarly, ptau levels were not significantly different in the Triton soluble fraction of 12 month S232D mice, yet the small sample size may have missed a small effect size with regard to ptau levels. However, the current data suggest that 14-3-3θ phosphorylation at S232 is not sufficient to induce αsyn or tau pathology at the time points examined.

Interestingly, while there was no change in 14-3-3θ levels at the earlier 3 month time point, there was a small but significant decrease in protein levels of 14-3-3θ at 12 months. As qPCR results showed no change in RNA levels between S232D and Cre control mice, this reduction in 14-3-3θ protein levels at 12 months is unlikely due to changes in transcription. Instead, 14-3-3θ phosphorylation may promote protein clearance and/or protein insolubilization. Indeed, insoluble 14-3-3θ phosphorylation levels correlated inversely with soluble 14-3-3 levels in human brains (McFerrin et al., 2017). With this age-related reduction, we predict that 14-3-3θ phosphorylation is likely to disrupt the ability of 14-3-3θ to protect against αsyn aggregation with age. Environmental or genetic triggers of αsyn pathology could induce more significant neurodegeneration in the presence of 14-3-3θ phosphorylation. Our findings here suggest that S232D expression alone may not be sufficient to induce

severe pathological changes *in vivo*, yet in the setting of another insult that induces neurodegeneration, phosphorylated 14-3-3θ may fail to adequately protect against misfolding of aggregation-prone proteins, such as αsyn. Consistent with this, we recently reported that S232D mice demonstrate increased αsyn inclusion formation upon αsyn PFF injection compared to Cre control mice (Wang et al., 2025).

The S232D mice showed behavioral deficits in the wire hang and social dominance tests. The wire hang test could be impacted by weight, and the S232D mice generally weighed more than the Cre control mice at the earlier ages but not at 12 months. We saw a persistent deficit in the wire hang in the S23D mice at 12 months, when weights were similar between both groups. We also used the HI to correct for weight differences, and the S232D mice still showed a wire hang deficit compared to Cre control mice at both 9 and 12 months of age.

The most pronounced behavioral difference that we observed was in the social dominance test. At all three time points tested, the S232D mice lost significantly more than the WT control mice. We and others have seen a decrease in social dominance in mice injected with PFF compared to αsyn monomer-injected mice (Stoyka et al., 2020; Underwood et al., 2021). We previously showed that 14-3-3θ overexpression can partially rescue the social dominance deficit observed in PFF-injected mice (Underwood et al., 2021). The finding that the S232D phosphomimetic mouse produces a similar effect to the PFF model indicates that the S232 phosphorylation observed in human disease could be a possible mechanism contributing to this behavioral deficit in an αsyn model.

We also performed subcellular fractionations in order to look at NMDAR levels at the PSD. We were interested in exploring this because NMDARs have previously been linked to social behavior in mice. In a water competition test of social dominance, mice with a mutation in *Grin1*, which encodes the NMDAR1, showed decreased social dominance (Ujita et al., 2018). The NMDAR blocker MK801 can interfere with winning in the social dominance tube test induced by photostimulation of dorsomedial prefrontal cortex neurons (Zhou

et al., 2017). Further, in a social preference test, vehicle treated mice spent more time in the social chamber whereas mice treated with ketamine, a NMDAR antagonist, spent more time in the non-social chamber (Mihara et al., 2017). Additionally, alterations in NMDARs have been reported in several neurological and neurodegenerative disorders (reviewed in Zhou and Sheng, 2013), and NMDAR agonists have been shown to be protective in both marmoset (Nash et al., 2000) and rat (Blandini et al., 2001) models of PD. Finally, a 14-3-3 functional knockout mouse, in which all 14-3-3 isoforms are inhibited by expression of the competitive peptide difopein, shows alterations in NMDARs at the PSD, a reduction in the NMDA/AMPA ratio in hippocampal CA1 recordings, and reduced evoked NMDAR synaptic currents in difopein mice (Qiao et al., 2014).

Our results regarding NMDAR levels in our S232D mice, however, were not straightforward. When NMDARs were normalized to PSD95, which we did initially to control for the purity of the fractionation, we saw significant increases in NMDAR2A and 2B. However, when we looked at PSD95 levels normalized to total protein on the blot, we saw a non-significant, though trending ($P$=0.0644), reduction in PSD95 levels suggesting that this 'increase' in NMDAR2A and 2B receptors in S232D mice may instead be due to changes in the density of the PSD. When normalized to total protein instead of PSD95, there were no significant changes in NMDAR1, 2A, and 2B between the S232D and Cre control mice. While further experiments, such as electron microscopy, are needed to confirm these findings, our experiments here seem to indicate that NMDAR levels are largely unchanged in S232D mice. Additionally, simply looking at receptor levels may not give the full picture of NMDAR function in these mice. Future studies are indicated to understand any electrophysiological changes involving NMDAR function that could potentially be occurring in the S232D mouse and contributing to the observed behavioral changes.

14-3-3 isoforms are known to heterodimerize, such that the S232D 14-3-3θ mutant could serve to act as a dominant negative for multiple 14-3-3 isoforms. The difopein (14-3-3 functional knockout) mouse reveals certain behavioral changes similar to those observed in S232D mice (Foote et al., 2015; Qiao et al., 2014). Neither the difopein nor the S232D mouse showed an anxiolytic phenotype, and both show changes in social behavior (Foote et al., 2015). Difopein mice show decreased preference for spending time with novel mice in either sociability or social novelty tasks (Foote et al., 2015) and showed impairment in the social dominance tube test, performing at the same level as wild-type mice injected with PFFs (Underwood et al., 2021). S232D mice similarly showed impairment in the social dominance tube test compared to controls here. However, the difopein mouse showed other extensive behavioral changes that were not observed in the S232D mouse, including cognitive deficits in fear conditioning, passive avoidance, and y-maze spontaneous alternation (Foote et al., 2015; Qiao et al., 2014). Additionally, we did not observe any dendritic changes in S232D hippocampal cultures, while difopein mice showed decreased dendritic complexity in both cortical layer V and hippocampal CA1 neurons *in vivo* (Foote et al., 2015). Examination of dendrites in more mature neuronal cultures or *in vivo* is indicated in future studies to test if S232D expression may affect dendritic architecture with age. The more pronounced behavioral and structural alterations in difopein mice compared to S232D mice suggests that the S232D 14-3-3θ mutant does not serve as dominant negative against all 14-3-3 isoforms.

There are several explanations for the limited behavioral and pathological findings observed in the S232D model. One possibility is that other 14-3-3 isoforms can compensate for any disruption in 14-3-3θ function. Another possibility is that more pronounced deficits in the S232 mouse may not be apparent until even older ages

than we tested here. Additionally, the S232D mutation may not fully mimic phosphorylation of 14-3-3θ. While replacing the serine with a bulkier aspartic acid can change the protein's interaction properties, it is not an identical structural change to a phosphate group. Similarly, the aspartic acid partially imitates the negative charge of a phosphate group, but not perfectly. Studies have shown that these phosphomimetics do not always completely mirror the properties of the phosphorylated protein (Corbit et al., 2003; Paleologou et al., 2008). However, current technology does not allow for manipulation of phosphorylation sites in mammalian *in vivo* models. Future technologies such as genetic code expansion may lead to better models to mimic phosphorylation *in vivo*. Comparison to the non-phosphorylatable mutant would be important to confirm our findings in the S232D mouse. We recently created a non-phosphorylatable S232A mutant, where the serine 232 is changed to an alanine (Wang et al., 2025). Repeating our behavioral and pathological experiments in the S232A mouse will be an important next step in understanding the physiology of the phosphorylation at S232 in 14-3-3θ.

Another limitation of our study is that expression of S232D is limited by expression driven by our Emx1-Cre driver. Our human brain studies have not revealed which cell type(s) are responsible for the higher levels of 14-3-3θ phosphorylation observed in PD, DLB, and AD. Crossing of the conditional S232D KI with other Cre driver lines could address whether certain brain cells are more susceptible to dysfunction related to 14-3-3θ phosphorylation.

Overall, the novel conditional KI S232 14-3-3θ phosphomimetic mouse showed subtle motor and social dominance deficits. These mice will be useful in understanding the role of 14-3-3 proteins in PD and other neurodegenerative disease models moving forward.

## MATERIALS AND METHODS
### S232D KI mouse
Mouse studies were done in compliance with the guidelines of the National Institute of Health (NIH) and University of Alabama at Birmingham (UAB) Institutional Animal Care and Use Committee (IACUC). UAB's IACUC approved these mouse studies.

As previously described, Cyagen created the conditional KI mouse expressing the 14-3-3θ S232D mutant (Wang et al., 2025). This conditional KI S232D mouse (Emx$^{+/+}$ S232$^{D/D}$) was then bred with the heterozygous Emx1-Cre mouse [B6.129S2-Emx1tm1(cre)Krj/J, The Jackson Laboratory strain #005628] (Emx$^{Cre/+}$ S232$^{WT/WT}$) (Gorski et al., 2002) to express the S232D mutant in cortical and hippocampal neurons. Heterozygous S232D mice (Emx$^{Cre/+}$ S232$^{WT/D}$) were then bred with Emx$^{+/+}$ S232$^{WT/D}$ mice to generate the groups of mice used in this paper: (1) WT mice (Emx$^{+/+}$ S232$^{WT/WT}$); (2) Cre control mice (Emx$^{Cre/+}$ S232$^{WT/WT}$); and (3) homozygous S232D KI mice (Emx$^{Cre/+}$ S232$^{D/D}$).

Genotypes were confirmed using Sanger sequencing of cDNA isolated from mouse cortices. RNA was isolated from homogenized cortex using the RNeasy Mini Kit (Qiagen, #74104) per manufacturer's instructions. Reverse transcription of RNA into cDNA was done using the Superscript IV Kit (Invitrogen, #18091050) per manufacturer's instructions. cDNA was submitted to the Heflin Center for Genomic Sciences core laboratory for Sanger sequencing.

### Behavior tests
#### Open field
Mice were placed in a 42 cm×42 cm clear plexi-glass box and allowed to freely move for 4 min. EthoVision software was used to record videos, which were analyzed for overall velocity and distance traveled.

#### Elevated plus/zero maze
Mice were placed in either a plus-shaped (elevated plus maze at 3 month time point) or circular-shaped (elevated zero maze at 9 and 12 month time points) maze with alternating open and closed segments. The mice were allowed to

wander freely for 4 min. The video recorded using EthoVision software was then analyzed to calculate the amount of time the mice spent in the open portions of the maze.

## Grip strength

Grip strength was measured using the Chatillon E-DFE-010 (cohort 1) or DFE-200 (cohort 2) Force Gauge. Mice were allowed to grip a mesh screen attached to the apparatus with their two front paws and were then gently pulled by their tails until they were pulled off of the screen. Mice were measured in five back-to-back trials, and the average of the trials was taken. Due to an instrument change between cohorts, grip strength data from the two cohorts were unable to be combined and are shown separately.

## Wire hang

Mice underwent a four-limb wire hang, as previously described (Klein et al., 2012; Underwood et al., 2021). Briefly, mice were placed on an apparatus in which a wire grid bottom is surrounded by angled sides to prevent the mouse from crawling over. The apparatus was hung on a ring stand, and mice were placed on top of the metal grid and allowed to securely grip the apparatus before it was flipped over with the mouse holding on upside down. The grid was stationed 24 inches above a cushioned rat cage. Latency to fall was recorded over two trials, which were averaged. All mice in a cohort completed one round before the next round began, resulting in an interval of approximately 1 h between trials. To account for weight differences, we calculated the HI, as described previously (Hoffman and Winder, 2016). Weight in grams was multiplied by the hang time in seconds to determine HI.

## Rotor rod

Mice were placed on a rotating rod apparatus (San Diego Instruments ROTOR-ROD™) that increased in speed from 0 to 35 RPM over 60 s for the 3 and 9 month time points and increased from 0 to 15 RPM for the 12 month time point. The latency to fall from the apparatus to a padded holding area approximately 24 inches below the rod was recorded, and mice were given a 4 min rest period between trials. Mice performed three trials per day for five consecutive days, with the first 3 days considered training days and the final 2 days used as testing days.

## Social dominance

The social dominance test was performed as previously described (Arrant et al., 2016; Stoyka et al., 2020; Underwood et al., 2021). Briefly, sex-matched mice were placed at opposite ends of a 12-inch-long clear plastic tube and allowed to enter the tube until they met in the middle. Mice were then observed until one mouse forced the other mouse out of the tube, and that mouse was recorded as the winner of that pair. Each mouse with paired against three different opponents, with cage mates avoided, and no mouse was run in back-to-back pairings. Depending on the size of the mice, either a tube of 1.5 inches or 1 inch diameter was used.

## Morris water maze

Morris water maze testing was done as previously described (Gannon et al., 2022). Mice were put in a pool with a hidden platform submerged just under the water surface. During the acquisition phase, four trials daily for 5 days were performed, and the time taken for the mice to find the platform (latency to platform) was determined. Mice started from a different starting point for each trial, and the order of the starting points changed each day. A probe trial was conducted in which the hidden platform was removed right after the final trial on day 5 of the acquisition phase. The time spent in the target quadrant where the platform had been located was measured. EthoVision software was used to record and analyze each trial.

## Passive avoidance

Passive avoidance testing was done as previously described (Gannon et al., 2022). A two-compartment box was used, in which one compartment was dark, and the floor in that compartment was composed of stainless steel bars connected to an internal shock source (Gemini II Avoidance System). On the training day, the mouse was placed into the lighted compartment with the door between the chambers closed. After 1 min, the door separating the compartments was opened, and the time it took for the mouse to cross to the

dark compartment was measured. Once in the dark compartment, the door was closed, and a 0.5 mA×2 s foot shock was administered. The mouse was left in the dark compartment for 30 s before being returned to their home cage. Mice were tested 24 h later with a similar protocol except without the foot shock, with a cutoff time of 5 min. The latency was measured as the time when all four paws entered into the dark compartment for both training and testing trials.

## Proteinase K digestion

Fourteen-month-old mice were perfused with PBS and then with 4% paraformaldehyde (PFA) using a forced pump system. 40 μm coronal sections through the striatum were mounted onto glass coverslips and dried overnight. The slides were then rehydrated with decreasing percentages of ethanol. The tissue was treated for 10 min with 50 μg/ml of proteinase K (Millipore Sigma, #706634) in TBS or TBS alone for control at 37°C. Slides were then washed in TBS and antigen retrieval was done in sodium citrate buffer (10 mM sodium citrate, 0.05% Tween-20, pH 6.0) for 1 h at 37°C. Sections were then permeabilized for 20 min using 0.25% Triton X-100 in TBS and were subsequently blocked in 5% normal goat serum with 0.1% Triton. Slides were next incubated overnight in primary antibody for αsyn (Cell Signaling Technology, #2628) at 4°C, washed in TBS, and then incubated in secondary antibody (Cy3-conjugated goat anti-rabbit, The Jackson Laboratory, #111-165-144) for 2 h at room temperature. After TBS washes, ProLong Diamond mounting media with DAPI (Thermo Fisher Scientific, #P36962) was used to mount the sections. Slides were imaged on an Olympus BX51 Microscope, and fluorescence intensity of striatal and cortical regions was calculated by a rater unaware of genotype and condition, using corpus callosum intensity for background subtraction.

## Immunofluorescence

40 μm coronal brain sections were incubated in antigen retrieval buffer (10 mM sodium citrate, 0.05% Tween-20, pH 6.0). After TBS washes, permeabilization with 0.1% Triton X-100, and blocking in 5% normal goat serum, sections were then incubated overnight in primary antibody for 14-3-3θ (Bethyl, #A303146A) and NeuN (EMB Millipore, # MAB377) or phospho S129 αsyn (EP1536Y) (Abcam, #ab51253). After further washes, sections were incubated at 4°C for 2 h in secondary antibody (Cy3-conjugated goat anti-rabbit, The Jackson Laboratory, #111-165-144, and Alexa Fluor 488 goat anti-mouse, Invitrogen, #A11029). After final washes, ProLong Diamond mounting medium with DAPI (Thermo Fisher Scientific, #P36962) was used to mount sections prior to imaging on an Olympus VS200 microscope.

## Triton and SDS fractionation

Mouse brains were homogenized and sonicated in lysis buffer containing 1% Triton X-100 as previously described (Wang et al., 2025). Following a 30 min incubation at 4°C, samples were centrifuged at 15,000 $g$ for 60 min. The Triton X-100 soluble fraction was the supernatant from this spin. Lysis buffer with 2% SDS was used to resuspend the pellet. Following a 10 s sonication, samples were again centrifuged at 15,000 $g$ for 10 min, and the supernatant was collected as the Triton X-100 insoluble sample.

## PSD fractionation

Subcellular fractionation was performed on mouse cortices following the protocol described by Bermejo et al. (2014) with minor modifications. Whole mouse cortices were homogenized using glass Teflon homogenizers in a 0.32 M sucrose solution in HEPES with protease and phosphatase inhibitors and then centrifuged at 900 $g$ for 10 min at 4°C (Eppendorf Centrifuge 5810 R). The supernatant (S1) was removed and was spun for 15 min at 10,000 $g$ at 4°C (BeckMan Optimal L-70K ultracentrifuge; SW 28 rotor). The resultant pellet (P2) was resuspended in the 0.32 M sucrose solution after removing the supernatant (S2) and then spun for an additional 15 min at 10,000 $g$ at 4°C (BeckMan Optimal L-70K ultracentrifuge; SW 28 rotor). This pellet (P2′) was resuspended in 1 ml of water with protease and phosphatase inhibitors and then rapidly homogenized with additional 3 ml of water using a glass Teflon homogenizer. After addition of HEPES, samples were gently rocked at 4°C for 30 min before being spun at 23,200 $g$ for 20 min at 4°C (BeckMan Optimal L-70K ultracentrifuge; SW 28 rotor). The pellet (P3) was resuspended in the 0.32 M sucrose solution and then layered on the top of a discontinuous sucrose gradient made up of 1.2 M sucrose,

1.0 M sucrose, and 0.8 M sucrose, all in 4 mM HEPES. The samples on the sucrose gradient were then centrifuged at 150,000 $g$ for 2 h at 4°C (BeckMan Optimal L-70K ultracentrifuge; SW 41 Ti rotor). Using a 16 G needle and 5 ml syringe, the tube containing the sucrose gradient was punctured slightly below the 1.0/1.2 M interphase, and the synaptic plasma membrane (SPM) layer was removed and placed in a 5 ml tube. 2.5 volumes of 4 mM HEPES were added to the sample and then spun at 200,00 $g$ for 30 min at 4°C (BeckMan Optimal XPN-100 ultracentrifuge; Type 70 Ti/70.1 Ti rotor). The supernatant was removed, and the pellet was resuspended in 50 mM HEPES, 2 mM EDTA. A portion was saved as the SPM and 0.54% Triton X-100 was added to the remaining amount. The samples were gently rocked at 4°C for 15 min and then spun at 31,100 $g$ for 20 min at 4°C (Eppendorf tabletop Centrifuge 5430 R). The supernatant was removed, and the pellet was resuspended in 50 mM HEPES, 2 mM EDTA and reserved as the PSD.

## Western blots

Samples were heated in 4× DTT sample loading buffer (0.25 M Tris-HCl, pH 6.8, 8% SDS, 200 mM DTT, 30% glycerol, Bromophenol Blue) at 100°C for 5 min and were loaded onto 15% SDS-polyacrylamide (PAGE) (Triton soluble and insoluble samples) or TGX Stain-Free FastCast Acrylamide (Bio-Rad, #1610185) (PSD fractions from subcellular fractionation) gels. 20 µg of each sample was loaded and run on the gel, and then transferred to 0.2 µm nitrocellulose membranes. Prior to transfer, the stain-free FastCast gels were activated using ChemiDoc MP Imaging System (Bio-Rad). Blots were then fixed in 0.4% PFA in TBS-T for 30 min and blocked for 1 h in either 5% milk in TBS-T (Triton soluble and insoluble samples) or Intercept (TBS) Blocking Buffer (Licor, #927-60001) (PSD fractions from subcellular fractionation). The membranes were incubated overnight at 4°C in primary antibody: total 14-3-3θ (5J20, Santa Cruz Biotechnology, #sc-69720), αsyn (BD Biosciences, #610787), phospho-tau (Thr181) (Cell Signaling Technology, #12885S), and GAPDH (Cell Signaling Technology, #2118S) (Triton soluble and insoluble samples); or NMDAR1 (Invitrogen, #320500), NMDAR2A (Novus, #NB300-105), NMDAR2B (Invitrogen, #MA1-2014), and PSD-95 (BioLegend 810401 clone K28/43) (PSD fractions). Primary antibodies were diluted in either 5% milk in TBS-T (Triton soluble and insoluble samples) or Intercept T20 (TBS) Antibody Diluent (Licor, # 927-65001) (PSD fractions). Blots were then washed in TBS-T and moved to secondary antibody for 1 h at room temperature (RT): HRP-conjugated goat anti-mouse IgG (Jackson ImmunoResearch, #115-035-146), HRP-conjugated goat anti-rabbit IgG (Jackson ImmunoResearch, #111-035-144), IRDye 680RD Goat anti-Mouse IgG (Licor, #926-68070), and IRDye 800CW Goat anti-Rabbit IgG (Licor, #926-32211). Blots using HRP secondaries were developed using Pierce ECL Western Blotting Substrate (Thermo Fisher Scientific, #32106) and then imaged using the ChemiDoc MP Imaging System (Bio-Rad). Data were analyzed using Image Lab 6.1 (Bio-Rad) and UN-SCAN-IT gel 6.1 (Silk Scientific). Blots using the Licor secondary antibodies were imaged using the Odyssey CLx (Licor). Data was analyzed using Image Studio (Licor).

## qPCR

RNA was extracted from cortices of 12 month old mice and reverse transcribed into cDNA, as described above. Primers against 14-3-3θ were used [5′-aggagtgacagcacacttgg-3′ (forward) and 5′-gttgcttctgaaaggaaacctc-3′ (reverse)] (Yacoubian et al., 2010). Actin was used to normalize PCR results [5′-tcctgaccgagcgtggctac-3′ (forward) and 5′-cggaaccgctcgttgccaat-3′ (reverse)]. 100 ng of cDNA was incubated with primers and PowerSYBR Green PCR Master Mix (Applied Biosystems, #4367659) for qPCR. Each sample was loaded in triplicate. Real-time qPCR was performed using a Bio-Rad CFX Opus 96, as previously described with slight modifications (Yacoubian et al., 2008). The following conditions were used: one cycle of denaturation at 95°C for 10 min; 40 cycles of denaturation at 95°C for 30 s, annealing at 58°C for 40 s, and polymerization at 72°C for 30 s. A melt curve was then calculated by performing 60 0.5°C increases in temperature, beginning at 65°C. Data were analyzed using the CFX Maestro software (Version 2.3; Bio-Rad).

For each primer, a standard curve was established using serially diluted cDNA from a WT mouse. cDNA starting concentration was quantified using a Nanodrop spectrophotometer (Fisher ND-2000C), and standards were serially diluted to concentrations of 256, 64, 16, 4, 1, and 0 ng/µl. Standard curves were generated for both 14-3-3θ and Actin primers using the known starting concentrations and the cycle threshold (Ct) for each standard sample. The approximate starting quantity of cDNA in each unknown sample was calculated relative to these curves using each sample's Ct value.

## Neurite analysis

Hippocampi from P0 mice were isolated and incubated with papain (Worthington Biochemical, #LK003176) for 20 min at 37°C, as previously described (Wang et al., 2018). After gentle dissociation using fire-polished glass pipettes and centrifugation at 1000 $g$ for 5 min, the cells were resuspended in Neurobasal-A medium (Gibco, #21103049) supplemented with B-27 (Gibco, #12587001) and 5% fetal bovine serum (FBS; Sigma, #F2442) and plated onto 18 mm glass coverslips pre-coated with poly-D-lysine (Sigma, #P0899). Following a 16-h incubation, the medium was replaced with Neurobasal-A supplemented with B-27 and 6 µM Arabinose C (LGC Standards, #EPC3350000).

At days in vitro (DIV) 8, primary neurons were fixed with 4% PFA and permeabilized with 0.5% Triton X-100. After blocking for 1 h in 5% normal goat serum in PBS, cells were incubated overnight at 4°C with a primary rabbit antibody against MAP2 (EMD Millipore, #AB5622). After PBS washes, the cells were exposed to a Cy3-conjugated goat anti-rabbit secondary antibody (Jackson ImmunoResearch, #111-165-144) for 2 h. The cells were then washed again with PBS before mounting the coverslips on slides using Prolong Diamond Antifade Mounting Media (Thermo Fisher Scientific, #P36962). Total dendritic lengths, Sholl analysis, number of nodes, and number of primary dendrites were quantified using Neurolucida analytical software (MBF Bioscience).

## Statistical analysis

GraphPad Prism 10.4.1 was used for statistical analysis of experiments. Data were analyzed using Student's unpaired $t$-test, or one-way or two-way ANOVA followed by post-hoc pairwise comparisons using either Šídák's or Tukey's multiple comparison tests. Significance was set at $P \leq 0.05$.

### Acknowledgements
We thank the UAB ADRC Neuropathology Core for the use of the Olympus VS200 microscope (P30AG086401).

### Competing interests
T.A.Y. has patent #7,919,262 issued to T.A.Y. The other authors declare no competing or financial interests.

### Author contributions
Conceptualization: M.A.G., T.Y.; Data curation: M.A.G., T.S., R.P., N.K., A.P., A.C.R., W.J.S., K.S., T.Y.; Formal analysis: M.A.G., T.S., R.P., N.K., A.P., A.C.R., T.Y.; Funding acquisition: T.Y.; Investigation: M.A.G., T.S., R.P., N.K., A.P., W.J.S., K.S., R.E.; Methodology: M.A.G., T.S., R.P., N.K., A.P., T.Y.; Project administration: T.Y.; Supervision: M.A.G., T.Y.; Validation: M.A.G.; Visualization: M.A.G., R.P., W.J.S., T.Y.; Writing – original draft: M.A.G., T.Y.; Writing – review & editing: M.A.G., T.S., R.P., N.K., A.P., A.C.R., W.J.S., K.S., R.E., T.Y.

### Funding
This work was supported by the National Institutes of Health (R01NS112203) and the Parkinson Association of Alabama. Open Access funding provided by University of Alabama at Birmingham. Deposited in PMC for immediate release.

### Data and resource availability
All relevant data can be found within the article and its supplementary information.

### Peer review history
The peer review history is available online at https://bio.biologists.org/lookup/doi/10.1242/bio.061963.reviewer-comments.pdf.

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
