## [Peer Review File · Biology Open]

A novel 14-3-3 θ phosphomimetic mouse model demonstrates social dominance defects

Thanushri Srikantha, Rudradip Pattanayak, Navya Kapa, Aneesh Pathak, A. Claire Roberts, William J. Stone, Kasandra Scholz, Roschongporn Ekkatine, Talene Yacoubian and Mary A. Gannon

DOI: 10.1242/bio.061963

Editor: Daniel Gorelick

Review timeline

Original submission:	26 February 2025
Editorial decision:	10 March 2025
First revision received:	5 May 2025
Editorial decision:	9 May 2025
Second revision received:	12 May 2025
Accepted:	13 May 2025

Original submission

First decision letter

MS ID#: bio.061963

MS TITLE: A novel 14-3-3 θ phosphomimetic mouse model demonstrates social dominance defects

AUTHORS: Thanushri Srikantha, Rudradip Pattanayak, Navya Kapa, Aneesh Pathak, A. Claire Roberts, William J. Stone, Kasandra Scholz, Roschongporn Ekkatine, Talene Yacoubian and Mary A. Gannon

I have now received both referees' reports on the above manuscript, and have reached a decision. The reviewer reports are shown at the bottom of this email or can be accessed, together with a copy of this decision letter, by going to:

As you will see, the referees express strong interest in your work. We would be happy to receive a revised manuscript addressing the reviewer's points.

With regards to reviewer #1, under major concerns, please pay close attention to comments 2 and 3. For comment 1, it does seem that including data from S232A mice would improve the completeness of the manuscript, however I would be open to arguments about why this would not be necessary for publication. Please address all of the minor concerns, especially minor concern 3. With regards to reviewer #2, you may ignore comment 4 (please do not remove any outliers from datasets, we value transparency).

If it would be helpful, you are welcome to contact us to discuss your revision in greater detail. Please send us a point-by-point response indicating your plans for addressing the referees comments, and we will look over this and provide further guidance.

At this stage, we also ask you to ensure your manuscript complies with our formatting guidelines. Provided you are able to fully address the referees' comments, we are positive about publication of your paper (we accept over 95% of revision submissions) and therefore hope you won't mind any extra work involved in reformatting your manuscript at this point.

Please ensure that you clearly highlight all changes made in the revised manuscript. Please avoid using 'Tracked changes' in Word files as these are lost in PDF conversion.

I should be grateful if you would also provide a point-by-point response detailing how you have dealt with the points raised by the reviewers in the 'Response to Reviewers' box. Please attend to all of the reviewers' comments. If you do not agree with any of their criticisms or suggestions please explain clearly why this is so.

Reviewer 1

Comments for the author

Gannon et al. investigated the effects of a phosphomimetic mutation (S232D) of 14-3-3 θ , on mouse behavior and on the levels of several proteins related to neurodegeneration and glutamatergic synapses. 14-3-3 θ is a brain-expressed protein that regulates θ \pm -synuclein folding, which plays a role in several neurodegenerative diseases, including Parkinson's disease (PD). Previous in vitro and in vivo studies have shown that overexpression of 14-3-3 θ protects neurons against PD-related pathology, and evidence suggests that phosphorylation at serine 232 (S232) inversely correlates with cognitive performance. S232 phosphorylation in the brain is increased in multiple neurodegenerative diseases, including PD and Alzheimer's disease.

The authors previously reported two conditional knock-in mouse lines to study S232 phosphorylation—a phosphomimetic mutation (S232D) and a phosphorylation-defective mutation (S232A) (Wang et al., 2025). In that study, cultured neurons from S232D mice formed more θ \pm -synuclein inclusions upon treatment with preformed θ \pm -syn fibrils than those from Cre control mice, while neurons from S232A mice formed fewer inclusions. Injection of preformed θ \pm -syn fibrils into the striatum induced more inclusions in the sensorimotor cortex of S232D mice compared to Cre controls.

In the present study, the authors used the S232D line to examine the effects of constitutive phosphorylation at this site on behavior and pathology. Specifically, they used the Emx1-Cre line to express the S232D mutation in the cortex and hippocampus. Behavioral tests included open field, elevated plus/zero maze, grip strength, wire hang, rotor rod, social dominance, Morris water maze, and passive avoidance. They found mild motor deficits and reduced social dominance in these mice, while cognition and anxiety levels were normal. These mice did not exhibit any θ \pm -synuclein or phospho-tau pathology, and dendritic arborization was normal in primary hippocampal cultures from these mice. Additionally, the levels of NMDA receptor 2A/B subunits in the postsynaptic density were increased.

The methods used to test the behavioral effects of the S232D mutation are appropriate, and the experiments include proper controls (though see below). Appropriate statistical tests were used, and overall, the study is rigorous.

Major Concerns:

1. It is curious why the authors did not test the S232A mutation using the same approaches. For the completeness of the work, including data from S232A mice seems essential, at least for the behavioral tests.
2. The authors conclude that pTau181 levels are not different between the Cre control and S232D groups based on Figure 5D. However, the two groups show a trend toward difference. With only 3 and 4 mice per group, the sample size does not seem sufficient for such a definitive conclusion. I suggest the authors tone down this conclusion.
3. The authors conclude that NMDAR2A/2B levels are increased in S232D mice based on results in Figure 6. The NR2A/2B levels were normalized to PSD95, but PSD95 levels were highly variable in this experiment, raising the question of whether PSD95 levels are affected by the S232D mutation. A control using a cellular protein, such as tubulin, should be used to normalize protein amounts in the samples.

Minor Concerns:

1. Figure 1C: Add a label to indicate the statistically significant difference at the 9-month time point.
2. Line 342: "nine" should be "9."

3. In this paper, the genotype labels use "+" to indicate the presence of a mutant allele or transgene and "-" for its absence. Although some investigators use this system, it can be confusing—for example, wild-type would be designated as "-/-." The suggestions from JAX below could help resolve this issue.

<https://www.jax.org/news-and-insights/jax-blog/2011/may/designating-genotypes-what-does-plus-really-mean20150422t150455>

Reviewer 2

Comments for the author

Gannon et al. investigated 14-3-3 θ phosphorylation using an S232D phosphomimetic knock-in mouse model. While 14-3-3 θ is neuroprotective in Parkinson's disease, its phosphorylation may contribute to neurodegeneration. S232D mice exhibited mild motor deficits and reduced social dominance but had normal cognition and anxiety. They showed no $\theta\pm$ -synuclein or phospho-tau pathology, and dendritic structure remained intact. However, increased NMDAR2A and NMDAR2B levels suggest altered NMDA signaling may underlie behavioral changes. Based on these findings, they concluded that this model serves as a valuable tool for studying 14-3-3 θ phosphorylation in neurodegeneration.

Overall, this is an interesting study that provides novel insight into the molecular mechanisms of PD. To improve the quality of the manuscript, the authors should address the following issues.

1. What is the underlying etiology caused by S232 phosphorylation of 14-3-3 θ ? What happens at the molecular or cellular level when 14-3-3 θ is phosphorylated at S232?
2. Have the authors assessed downstream pathways affected by S232 phosphorylation? Some discussion about the downstream effects of S232 phosphorylation in the context of PD would be helpful.
3. Figure 7 -The authors state that S232D mice showed increased levels of NMDAR2A and NMDAR2B. Are there any reports indicating that increased NMDAR2 levels are associated with neurodegenerative disorders?
4. Figure 2 - In the wire hang test, one Cre+/- S232D-/- mouse at 9 months exhibited an extremely long hanging time (over 600 seconds). This outlier should be considered for removal.
5. In the rotarod test, both Cre+/- S232D-/+ and Cre+/- S232D+/+ mice had times of less than 20 seconds. These results seem unusually short for controls. It should be carefully evaluated whether these mice truly exhibit motor deficits.
6. Figure 5A - The authors state that S232D mice did not show $\theta\pm$ -synuclein ($\theta\pm$ Syn) pathology. However, only an anti- $\theta\pm$ Syn antibody was used in immunohistochemistry. To confirm the presence or absence of aggregates, it would be better to include an anti-phosphorylated $\theta\pm$ Syn antibody.
7. Figure 5D, E - The authors report a decrease in 14-3-3 θ protein levels. What is the cause of this reduction? Does the decrease result from reduced mRNA expression, or is it due to increased protein turnover? Further clarification is needed.

Reviewer's Responses to Questions

Experimental quality

Does each figure have the proper controls?

If 'No', please indicate reasons in Comments for Author box below.

Reviewer #1:

Yes

Reviewer #2:

Yes

Were the data analyzed using appropriate statistical tests?

If 'No', please indicate reasons in Comments for Author box below.

Reviewer #1:

Yes

Reviewer #2:

Yes

Reproducibility

Were experiments performed using adequate number of biological replicates?
If 'No', please indicate reasons in Comments for Author box below.

Reviewer #1:

Yes

Reviewer #2:

Yes

Does the methods section provide sufficient detail to permit reproducibility?
If 'No', please indicate reasons in Comments for Author box below.

Reviewer #1:

Yes

Reviewer #2:

Yes

Completeness

Are the manuscript's conclusions supported by the data?
If 'No', please indicate reasons in Comments for Author box below.

Reviewer #1:

No

Reviewer #2:

Yes

Scholarship

Do the authors cite and discuss the merits of data that would argue for and against their conclusion?
If 'No', please indicate reasons in Comments for Author box below.

Reviewer #1:

Yes

Reviewer #2:

Yes

Does the manuscript title & abstract accurately reflect the contents of the manuscript, without hyperbole?

If 'No', please indicate reasons in Comments for Author box below.

Reviewer #1:

Yes

Reviewer #2:

Yes

First revision

Author response to reviewers' comments

Dr. Daniel A Gorelick

Biology Open Handling Editor

Dear Dr. Gorelick,

Enclosed please find a revised copy of the manuscript “A novel 14-3-3 θ phosphomimetic mouse model demonstrates social dominance defects” by Mary Gannon et al. that we are resubmitting for publication in Biology Open. All authors have read and approved this manuscript.

We are very appreciative of the comments from the reviewers and believe that addressing their concerns has improved the quality of our manuscript. A point-by-point response to the reviewers' concerns is below, and has also been uploaded on the 'attach files' page:

1. It came to our attention that one of the mice used in the PK digestion experiment in Fig. 5A was the incorrect genotype. We have excluded this mouse from the analysis and updated the figure and the statistics.

Reviewer 1:

1. We agree that behavioral and morphological evaluation of our new non-phosphorylatable mutant, S232A, mice would be an interesting addition to the manuscript, yet this evaluation has not yet commenced for several reasons. When we began this project, we had not yet created the S232A mouse line and therefore performed our year-long behavioral analysis in the S232D line only. The S232A mice are a new addition in our lab and have so far been used in primary culture-based experiments. To complete the longitudinal behavioral studies with the S232A mice that we did with the S232D mice would likely require close to two years by the time a cohort of mice could be bred, run through 12-14 months of behavior, and then euthanized and the tissue processed. Additionally, we are still awaiting funding of a NIH R01 grant (which got a fundable score in 2024) to pursue further experiments using the S232A and S232D mice. Unfortunately, we do not yet have the funds to pursue any behavioral assessment of the S232A mice.

2. Reviewer 1 has noted that the relatively small sample size for the ptau western blots can make it difficult to draw definitive conclusions, particularly when the data could be trending towards a

decrease in the S232D mice. We have made note of these limitations of this experiment in the Discussion on page 21 of the manuscript.

3. Reviewer 1 expressed concerns over normalizing the NMDAR western blots to PSD95, which had variable expression in the blots (Fig. 7), and suggested using a different loading control to normalize. As these blots were run on total protein gels, we checked the PSD95 levels normalizing to total protein and found that PSD95 levels trended lower in the S232D mice compared to Cre controls (new Fig. 7E) - suggesting that there could be a reduction in the post-synaptic density in S232D mice. Because of that difference, we normalized NMDAR levels to total protein instead of PSD95 and found that NMDAR to total protein levels did not differ (new Fig. 7F-H). We have added in panels E-H in Figure 7 and have updated the manuscript with the results and statistics on pages 19-20. We have also updated the discussion, removing references to increases in NMDAR2A and 2B, and updated the title to remove mention of the NMDAR changes.

4. We have added a p value to the nine month rotor rod graph in Fig. 1C to indicate the significant interaction effect as requested by Reviewer 1.

5. Reviewer 1 notes that "nine" should be "9" on line 342. We followed the general APA number guidelines that suggest writing out numbers through ten and using numerals for over ten throughout the manuscript, and therefore we did not make this change in order to be consistent.

6. We understand that our nomenclature for the mutant mice may have been confusing to follow. Using the guidelines suggested by Reviewer 1, we have tried to make the naming more straightforward. We have made the following changes for the experimental mice used:

- a. WT control: previously Emx1-Cre $-/-$ S232D $-/-$; now Emx $+/+$ S232WT/WT
- b. Cre control: previously Emx1-Cre $+/-$ S232D $-/-$; now EmxCre/ $+$ S232WT/WT
- c. S232D: previously Emx1-Cre $+/-$ S232D $+/+$; now EmxCre/ $+$ S232D/D

We also updated all of our figure labels to use just the WT control, Cre control, and S232D naming that matches how the mice are referred to in the text.

Reviewer 2:

1. Reviewer 2 inquired as to what causes S232 phosphorylation and what happens at the molecular or cellular level with S232 phosphorylation. We added a brief discussion of some of the work showing that pesticides such as rotenone can induce S232 phosphorylation (page 20). Additionally, we refer to our molecular modeling work which points to changes in the binding pocket of 14-3-3 θ upon S232 phosphorylation (page 21).

2. Reviewer 2 asked to add some discussion regarding the potential downstream effects of S232 phosphorylation. We have added a brief discussion of our work looking at the effects of S232 phosphorylation on both α syn and LRRK2 (pages 20-21) and have recently submitted a manuscript looking at changes in 14-3-3 θ 's interactome induced by S232 phosphorylation.

3. Given that we initially saw changes in NMDAR receptor subunits, Reviewer 2 asked for a discussion of NMDAR changes that have been reported in neurodegenerative disorders. While the differences we had initially seen were not present when re-normalized, we still added a brief statement acknowledging that changes in NMDARs have been noted in neurological diseases (pages 22-23).

4. Reviewer 2 pointed out that there is a data point in the wire hang data that is higher than the rest of the data set, and suggested removing it if it was an outlier. However, the editor has requested that no data points be removed, and therefore we have left this unchanged.

5. Reviewer 2 noted that the mice, as a whole, did not stay on the rotarod for very long and therefore care should be taken in concluding that there is actually a motor deficit. The conclusion of mild motor impairment is based on the wire hang data, which shows much more prominent effects. Further, while we do report a significant interaction effect at the nine month timepoint only, we do not conclude that there is an appreciable change in the rotarod performance. We believe the short time that the mice stayed on the rotarod may be an effect of the equipment that was used. We performed this experiment on a rotarod made by San Diego Instruments. This instrument has since been replaced by a new one by Ugo Basile, and we have noticed that there are several differences between the two. The speed settings are different, the circumference of the rotating rod is different, and the height above the ground of the rod is different. The mice seem able to stay on the new equipment for longer. However, we are confident in our data, as all of the mice were run on the same San Diego Instruments equipment for this experiment. We have also previously reported rotarod data using the San Diego Instruments rotarod where the mice all had very low latencies to fall (Fig. S2 in Underwood, R., Gannon, M., Pathak, A. et al. 14-3-3 mitigates alpha-synuclein aggregation and toxicity in the in vivo preformed fibril model. *Acta Neuropathol Commun* 9, 13 (2021) <https://doi.org/10.1186/s40478-020-01110-5>).

6. Reviewer 2 recommended using an anti-phosphorylated α syn antibody. We have added this experiment and confirmed the absence of α syn aggregates detectable by a pS129 phosphorylated α syn antibody in S232D mice (Supplemental Fig. 3). We included a positive control brain that had been injected with preformed fibrils to ensure that the lack of p α syn signal in the S232D and Cre control brains was not due to an experimental issue.

7. Reviewer 2 asked for clarification on whether the decrease in 14-3-3 θ seen in the 12 month brain tissue was due to decreased mRNA expression or increased protein turnover. To answer this question, we performed qPCR on 12 month cortical lysates. We did not see a difference in the qPCR and have added the quantification in Fig. 5B and included a discussion of the results on page 21.

We hope that the changes and clarifications that we have made will satisfactorily address the reviewers' concerns and that we have met the requirements for publication in Biology Open.

Second decision letter

MS ID#: bio.061963R1

MS TITLE: A novel 14-3-3 θ phosphomimetic mouse model demonstrates social dominance defects

AUTHORS: Thanushri Srikantha, Rudradip Pattanayak, Navya Kapa, Aneesh Pathak, A. Claire Roberts, William J. Stone, Kasandra Scholz, Roschongporn Ekkatine, Talene Yacoubian and Mary A. Gannon

Thank you for submitting a revised manuscript and fully addressing the reviewer comments. The additional experiments you performed, and the changes you made to the text, have improved the rigor of your manuscript. The complex mouse genotypes are now clearly explained using nomenclature that is intuitive to our readers (thank you!). If you wouldn't mind addressing the following minor comments, which do not require any additional experiments, only small changes to the text, then we will gladly publish your manuscript:

In the methods section describing the rotor rod, please specify the model rotor rod that you used and the manufacturer.

For the qPCR methods, could you please describe how you got from melt curves to changes in relative gene expression? $\Delta\Delta$ Ct method? other? If you could provide sufficient details as described in the MIQE guidelines (PMID 19246619) that would be wonderful for transparency purposes.

Your reasons for not including results from S232A mice are understandable. Would you mind adding a note to the Discussion saying something to the effect that future studies should look at the unable-to-be-phosphorylated S232A mutant? I don't want to give away your ideas, but I do think (as a reviewer mentioned) that this is an obvious experiment to do and so acknowledging that would improve the discussion. You certainly don't need to mention the details about why you weren't able to analyze these mice for this manuscript, although I personally understand and empathize with you.

Did the PSD95 levels change in S232D mice versus controls? Your data indicate that there was no statistically significant difference between PSD95 levels in S232D vs control. Which would then suggest that there was an increase in NMDAR2A and NMDAR2B proteins in S232D vs controls, when normalized to PSD95. To make this more clear, can you add a bar labeled NS (or similar) to indicate no statistically significant difference between S232D and control mice in Fig 7 panels E-H? I loved the way you handled this in the discussion section (line 614-617), although I might clarify the wording - are you defining "potential reduction" as a reduction in the mean, albeit a reduction that is not statistically significant as defined by $p < 0.05$? If you could make this more clear, I'd be grateful.

At this stage, we also ask you to ensure your manuscript complies with our formatting guidelines " please see our manuscript preparation guidelines for details. Provided you are able to fully address the referees' comments, we are positive about publication of your paper (we accept over 95% of

revision submissions) and therefore hope you won't mind any extra work involved in reformatting your manuscript at this point.

Please ensure that you clearly highlight all changes made in the revised manuscript.

Please avoid using 'Tracked changes' in Word files as these are lost in PDF conversion.

I should be grateful if you would also provide a point-by-point response detailing how you have dealt with the points raised by the reviewers in the 'Response to Reviewers' box. Please attend to all of the reviewers' comments. If you do not agree with any of their criticisms or suggestions please explain clearly why this is so.

Second revision

Author response to Editor comments

We are very happy to hear that our manuscript will be accepted for publication with minimal changes. We have made the requested changes, as outlined below:

1. We have added the manufacturer and model of the rotor rod apparatus into the methods section of the manuscript (page 7, lines 150-151).
 2. We have updated the qPCR methods, using the MIQE guidelines (pages 11-12, lines 280-301).
 3. We added in a short statement indicating future directions to repeat these experiments in our non-phosphorylatable S232A mouse (pages 24-25, lines 672-677).
 4. We clarified the change that we see in PSD95 levels in the S232D and control mice. We updated the text on page 23, line 629 from 'potential' to 'non-significant, though trending ($p=0.0644$)', and also added the p-value to the graph in Figure 7E.
-

Third decision letter

MS ID#: bio.061963R2

MS TITLE: A novel 14-3-3 θ phosphomimetic mouse model demonstrates social dominance defects

AUTHORS: Thanushri Srikantha, Rudradip Pattanayak, Navya Kapa, Aneesh Pathak, A. Claire Roberts, William J. Stone, Kasandra Scholz, Roschongporn Ekkatine, Talene Yacoubian and Mary A. Gannon

Thank for making the minor revisions as requested. I am happy to tell you that your manuscript has been accepted for publication in Biology Open, pending our standard publication integrity checks.